



# Morphological controls on Hortonian surface runoff:
# An interpretation of steady-state energy patterns, maximum power states and dissipation regimes within a thermodynamic framework

Samuel Schroers[1], Olivier Eiff[2], Axel Kleidon[3], Ulrike Scherer[4], Jan. Wienhöfer[1], Erwin Zehe[1]

[1]Institute of Water Resources and River Basin Management, Karlsruhe Institute of Technology – KIT, Karlsruhe, Germany
[2]Institute for Hydromechanics, Karlsruhe Institute of Technology – KIT, Karlsruhe, Germany
[3]Max-Planck Institute for Biochemistry, Hans Knöll Str. 10, 07745 Jena, Germany
[4]Engler-Bunte-Institut, Water Chemistry and Water Technology – KIT, Karlsruhe, Germany

*Correspondence to*: S. Schroers (samuel.schroers@kit.edu)

**Abstract.** Recent developments in hydrology have led to a new perspective on runoff processes, extending beyond the classical mass dynamics of water in a catchment. For instance, stream flow has been analysed in a thermodynamic framework, which allows the incorporation of two additional physical laws and enhances our understanding of catchments as open environmental systems. Related investigations suggested that energetic extremal principles might constrain hydrological processes, because the latter are associated with conversions and dissipation of free energy. Here we expand this thermodynamic perspective by

exploring how hillslope structures at the macro- and microscale control the free energy balance of Hortonian overland flow. We put special emphasis on the transitions of surface runoff processes at the hillslope scale, as hillslopes energetically behave distinctly different in comparison to fluvial systems. To this end, we develop a general theory of surface runoff and of the related conversion of geopotential energy gradients into other forms of energy, particularly kinetic energy as the driver of erosion and sediment transport. We then use this framework at a macroscopic scale to analyse how combinations of typical

hillslopes profiles and width distributions control the spatial patterns of steady-state stream power and energy dissipation along the flow path. At the microscale, we analyse flow concentration in rills and its influence on the distribution of energy and dissipation in space. Therefore, we develop a new numerical method for the Catflow model, which allows a dynamical separation of Hortonian surface runoff between a rill- and a sheet flow domain. We calibrated the new Catflow-Rill model to rainfall simulation experiments and observed overland flow in the Weiherbach catchment and found evidence that flow

accumulation in rills serves as a means to redistribute energy gradients in space, therefore minimizing energy expenditure along the flow path, while also maximizing overall power of the system. Our results indicate that laminar sheet flow and turbulent rill flow on hillslopes develop to a dynamic equilibrium that corresponds to a maximum power state, and that the transition of flow from one domain into the other is marked by an energy maximum in space.

## 1 Introduction

Runoff is of key importance to biological, chemical, and geomorphological processes. Landscapes, habitats, and their functionalities are coupled to the short and long-term evolution of rainfall-runoff systems. As we live in a changing environment it has been of mayor interest to explain the development of runoff systems and how ecological (Zehe et al., 2010; Bejan and Lorente, 2010), chemical (Zhang and Savenije, 2018; Zehe et al., 2013) and geomorphological (Leopold and Langbein, 1962; Yang, 1971; Kleidon et al., 2013) processes organize in time and space in order to deplete the free energy

provided by water flow. In this study we direct our focus on the most apparent runoff, stream flow in river systems and subsequently surface runoff on hillslopes. Typically, surface runoff and its momentum balance are characterized by friction laws such as they are expressed by Dary-Weißbach, Manning or Chezy (Nearing et al., 2017). Consequently, the estimations of hydraulic flow are bound to semi-empirical parameters which in essence express the ability of a system to dissipate free energy to bound thermal energy (Zehe and Sivapalan., 2009). The rate at which dissipation happens is determined by these



parameters, which are primarily assumed to be constant coefficients for given hydraulic conditions but often are used as calibration parameters in order to reproduce the observed values of a variable, such as flow velocities or water levels. Our understanding of the development of complex hydrological systems, which include feedback loops between flow and its driving energy gradients is therefore incomplete if we do not include an additional principle or law that sheds light on dissipation (Paik and Kumar, 2010, Singh, 2003).

Leopold and Langbein (1962) were amongst the first to introduce thermodynamic principles in landscape evolution. Representing a one-dimensional river profile as a sequence of heat engines with pony brakes, they showed that the most likely distribution of potential energy per unit flow along a rivers course follows an exponential function. Their main hypothesis is the principle of least work, or equivalently, the constant production of entropy per flow volume. Yang (1976) extended this principle and termed it minimum stream power. He effectively detailed how flow velocity, slope, depth and channel roughness

of a stream should adjust in order to fulfil the hypothesis of minimum steam power. In his work about optimal stream junction angles, Howard (1990) also assumed that stream power is minimized, while Rodriguez-Iturbe et al. (1992) in their study about optimal drainage patterns deduced that drainage networks minimize overall energy dissipation. Therefore, the authors considered three principles, which were also applied and tested in subsequent papers (Rodriguez-Iturbe et al., 1994; Ijjaz Vasquez et al., 1993). These have been termed (1) the principle of minimum energy expenditure in any link of the network,

(2) the principle of equal energy expenditure per unit area and (3) the principle of minimum total energy expenditure in the network as a whole. Notably these authors were able to show that by application of these principles they could produce three-dimensional drainage networks that follow geometric laws observed in nature (Horton's law of stream number and Horton's law of stream lengths; Horton 1945).

Closely related is the principle of maximum entropy production (Paltridge, 1979) which states that driving energy gradients

are depleted as fast as possible. This implies a maximization of the product of flow and driving gradient, or stated differently, a maximization of the rate of free energy production und thus power (Kleidon et al., 2013). Kleidon et al. (2013) argue that while the driving geopotential gradient is depleted at the maximum rate, the associated sediment export maximizes with the same rate. The authors also analyse how the formation of structures (channels) enhances these dynamics and conclude that maximum power is in line with the preceding principle of minimization of frictional losses as transfer of energy to sediment

maximizes and frictional dissipation of kinetic energy minimizes simultaneously.

Motivated by their similarity to river networks, several studies tested whether hillslope scale rill networks develop in accordance with the minimum energy expenditure theory of river systems (Gómez et al., 2003; Rieke-Zapp and Nearing, 2005; Berger et al., 2010). Indeed, Rieke-Zapp and Nearing (2005) as well as Gomez et al. (2003) found in their experiments that the development of rill networks on hillslope scale follows the same trend of minimization of energy expenditure as it was

proposed for river networks (Ijjasz-Vaquez et al., 1993). However, in contrast to the analysis of the development of pure networks and their self-similar characteristics (Rodriguez-Iturbe et al., 1994), hillslope-scale analysis of thermodynamic principles must include the transitional emergence of rills as they do not exist a priori (Favis-Motlock et al., 2000). The common assumption here is that rill flow reduces the volume specific dissipative loss due to a larger hydraulic radius (Berkowitz and Zehe, 2020), which causes larger flow velocities compared to sheet flow. However, this assumption was not

validated. The geometry and topology of these drainage networks are time-dependent, as a response to transient flows of water and sediments, and these networks develop in a self-reinforcing manner (Gómez et al., 2003; Rieke-Zapp and Nearing, 2005; Berger et al., 2010).

Micro rills emerge at some critical downstream distance on the hillslope (cf. Horton's (1945) "belt of no erosion") and usually continue in parallel for some length before they merge into larger rills (Schumm et al., 1984). Sometimes they split apart and

can expand into larger gullies (Achten et al., 2008; Faulkner, 2008) before finally forming the river channel. This transitional emergence of a structured drainage network was firstly stated in Playfair's Law (cited in Horton, 1945) and has since then been observed in a variety of studies (Emmett, 1970; Abrahams et al., 1994; Evans et al., 1995).



The optimization of hydraulic geometry or structure through erosion and deposition of soil and sediment is but one mechanism which works in thermodynamic rainfall-runoff systems. In some of the most comprehensive field and laboratory experiments which were dedicated to investigating hydraulics of overland flow, Emmett (1970) measured a transition from laminar to turbulent flow on hillslope plots of up to 12m downstream length. This suggests that the upslope laminar runoff dissipates less energy per unit volume and further downstream evolves to a more dissipative turbulent flow regime. Parsons et al. (1990) measured overland flow hydraulic conditions on a semiarid hillslope in Arizona and attributed an observed downslope decrease of frictional flow resistance to the accumulation of flow in fewer rills, like a transition of inter-rill flow, from here onwards referred to as sheet flow (Dunne and Dietrich, 1980), to rill flow. More recently a concept emerged that upholds a theory of slope-velocity equilibrium (Nearing et al., 2005) on hillslopes, proclaiming that physical and therefore hydraulic roughness adapts so that flow velocity is a unique function of overland flow rate independent of slope. All in all, these studies are a concise selection of principles and hypotheses which have been published in the past about the organization of overland flow on hillslopes and river systems but reflect the overall notion that hydrological systems evolve towards a meta-stable energetically optimal configuration (Zehe et al., 2013; Kleidon et al., 2014, Bejan and Lorente, 2010).

Recently, Kleidon (2016) presented a framework which can be applied to each of these principles and beyond. In essence, the mentioned studies investigated different aspects of energy depletion provided by nature, suggesting that energy and energetic flow is organized in time and space. On different scales different mechanisms are involved to a higher or lesser extent but the applicability of thermodynamic laws does not change. Budgeting of energy conversion not only allows the incorporation of all types of free energy but might also help explain why some conversion processes prevail at certain scales. Therefore, our goal is twofold. First, we present the general thermodynamic framework and how surface runoff in river systems and on hillslopes fit into this setting. We propose that despite the similarity of hillslope and river runoff, their energetic functioning is distinctly different. This is because river elements are mainly fed from the upstream discharge (Kleidon et al., 2013) while hillslope elements receive substantial water masses through rainfall input and upslope runon. We will show that the latter causes a trade-off in the potential energy of overland flow as an increasing mass of water flows along a continuously declining geopotential. We hypothesize that these antagonistic effects imply a distinct point in space with maximum potential energy of overland flow. This local maximum is a hot spot as it separates upslope area where the flow gains potential energy from a downslope part, where the hillslope behaves energetically like a river with declining potential energy. This relates to our second hypothesis, that the build-up of energy happens under laminar flow conditions with less dissipation per unit flow rate, whereas more dissipative, turbulent flow should dominate if total potential energy is declining. For steady state energy conversion processes, production of entropy is represented by the distribution of energy gradients in space, and any system would have to optimize these gradients to behave in accordance with any of the mentioned optimization principles. Here we suggest that this optimization happens at several scales and therefore involves different dominant flow processes.

The first application of our thermodynamic framework, presented in sect. 2, zooms on the principal mechanisms involved in steady state energy distribution and therefore gradients of surface runoff on hillslopes by exploring macro scale -topographic controls. We explore how different distributions of geopotential along the flow line (cf. Rieke-Zapp and Nearing, 2000) in combination with different hillslope width functions control flow accumulation and thus overland flow energy in space (sect. 3). In the second part of the study (sect. 4), we apply our framework to observations of overland flow distinguishing rill- and sheet flow obtained during rainfall-runoff simulation experiments in the Weiherbach catchment (Scherer et al., 2012). For this we develop a rill domain for the numerical model Catflow (Zehe et al., 2001), simulate distinctly different experiments and analyse them in terms of overland flow energy and the transition from laminar to turbulent flow. The separation of surface runoff into sheet- and rill flow allows us to explore the effect of runoff accumulation in rills on energetics of overland flow and test our second hypothesis that the energy maximum coincides with the transition from laminar to turbulent flow regime.



## 2 Theory

### 2.1 Free energy balance of rainfall-runoff systems

We start very generally with the first law of thermodynamics, to express energy conservation of surface runoff in the following form:

$$\frac{dU}{dt} = \frac{d(H)}{dt} + \frac{dW}{dt} \qquad (1)$$

which states that a change in the internal energy U [Joule] of a system consists of the transfer of heat H [Joule] to the system plus the amount of work W [Joule] performed by the system. Here, we add performed work to the internal energy, as in an open environmental system the amount of energy $dW$ does not leave the system but rather is converted into some other kind of energy that stays inside the system (Kleidon, 2016). Note that the capacity of a system to perform work is equivalent to the term "free energy", whereas heat is associated with the dissipation of free energy and production of thermal entropy. The latter reflects the second law of thermodynamics, which states that entropy is produced during irreversible processes. The free energy of surface runoff at any point on the hillslope corresponds to the sum of its potential and kinetic energy if we neglect pressure work (i.e., assuming constant pressure), mechanical work (i.e., no shaft work such as pumps and turbines) and chemical energy. We apply Eq. (1) for each energy form, meaning that the difference between in- and outflux of energy causes a gradient, that can be depleted by conversion of energy into another form. For potential energy we consider the part which corresponds to the topographic difference between precipitation input and runoff output as available potential energy, which is fed by an influx of free energy from precipitation and upslope runon. Potential energy of infiltration excess water at the hillslope surface is converted into kinetic energy of overland flow, and kinetic energy is dissipated into heat (Fig. 1). In this two-box scheme we consider only the energies of fluid flow and neglect energy transfer from and to sediment particles.

To explore the spatial distribution of energy we subdivide the hillslope surface into segments along the horizontal flow path $x$ (Fig. 2) with given width $B$ and express energy fluxes in W m$^{-1}$. We can thus write the energy balance equations for any segment of the hillslope:

$$\frac{dE_f^{pe}(x)}{dt} = J_{f,in}^{pe}(x) - J_{f,out}^{pe}(x) + J_{P,in}^{pe}(x) - J_{Inf,out}^{pe}(x) - P_f(x)$$
$$= J_{f,net}^{pe}(x) + J_{Peff,net}^{pe}(x) - P_f(x) \qquad (2)$$

$$\frac{dE_f^{ke}(x)}{dt} = P_f(x) - D_f(x) + J_{f,in}^{ke}(x) - J_{f,out}^{ke}(x) = P_f(x) - D_f(x) + J_{net}^{ke}(x) \qquad (3)$$

Fluxes with superscript "pe/ke" relate to potential energy and kinetic energy, respectively. The subscript "f" relates to surface runon and runoff, subscript "inf" to infiltration and subscript "P" to precipitation (see table 1). Eq. (2) balances changes of potential energy of runoff $E_f^{pe}$, while Eq. (3) is the kinetic energy balance of surface water $E_f^{ke}$. We further define $J_{f,net}^{pe}, J_{f,net}^{ke}, J_{Peff}^{pe}$, as the net energy fluxes across the systems boundary. All energy fluxes are in watt per meter (Table 1), and the subscript "net" represents the difference of the respective "in" and the "out" fluxes. $P_f$ is the transfer from potential to kinetic energy and $D_f$ is the remaining energy flux, which has not been conserved as potential or kinetic energy of the water flow and leaves the system. $D_f$ is more than dissipation by friction, as it includes e.g., work needed for sediment detachment and transport. While dissipation means free energy is lost as heat, kinetic energy transfer to the sediment is not dissipated, as it creates macroscopic motion. In the following, we neglect the kinetic energy transfer to sediments (and other mass) and refer to D$_f$ simply as the dissipation for of potential energy. As generally accepted, we assume that infiltration and precipitation act mostly on the potential energy and neglect their influences on kinetic energy.



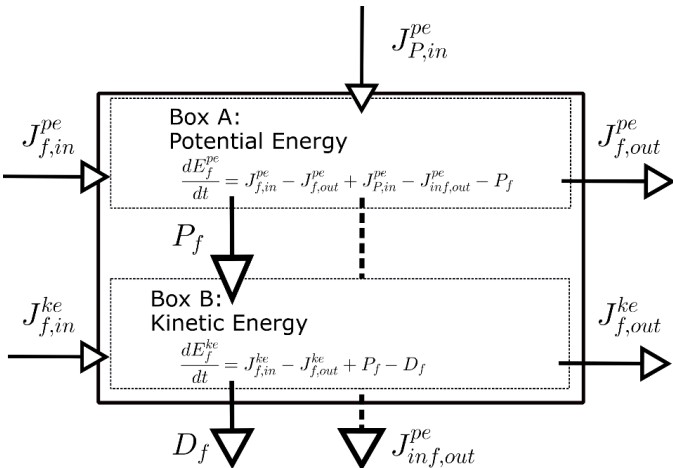

**Figure 1: The hillslope surface runoff as a two box open thermodynamic system**

Combining Eq. (2) and (3) and accounting time dimensionality, the total free energy balance is:

$$\frac{dE_f^{pe}(x,t)}{dt} + \frac{dE_f^{ke}(x,t)}{dt} = J_{f,net}^{pe}(x,t) + J_{f,net}^{ke}(x,t) + J_{Peff,net}^{pe}(x,t) - D_f(x,t) \tag{4}$$

The change in total free energy of the overland flow system is the sum of the differences in the net boundary energy fluxes minus dissipation. In the case of steady state $\left(\frac{dE_f^{pe}(x,t)}{dt} = \frac{dE_f^{ke}(x,t)}{dt} = 0\right)$, dissipation $D_f$ can be maintained if the net boundary fluxes are non-zero. It should be noted that within the balance of total free energy the net boundary fluxes of a kind may become negative, therefore seemingly the system exports more energy than is imported. Downslope water movement does not

just imply a reduction in its geopotential, but also that additional rainfall-mass is added on its way. In such a distributed mass accumulating system, it is therefore possible that the gain in mass per unit length adds more potential energy than is converted into kinetic energy. From a momentum balance point of view this corresponds to an increase in momentum while the velocity is constant (Kleidon et al., 2013).

**Table 1: Overview of the different symbols used in this study**

| symbol | unit | description |
|---|---|---|
| $U$ | [kg m² s⁻²] | internal energy of a thermodynamic system |
| $W$ | [kg m² s⁻²] | available energy to perform work by the thermodynamic system |
| $H$ | [kg m² s⁻²] | thermal energy of the thermodynamic system |
| $E_f^{pe/ke}$ | [kg m s⁻²] | Potential- or kinetic energy of the water flow |
| $J_{f,in/out}^{pe/ke}$ | [kg m s⁻³] | Potential- or kinetic energy flux entering or leaving the system |
| $J_{P,in}^{pe}$ | [kg m s⁻³] | precipitation entering the system as potential energy flux |
| $J_{inf,out}^{pe}$ | [kg m s⁻³] | infiltration leaving the system as potential energy flux |
| $P_f$ | [kg m s⁻³] | power to create kinetic energy of system |
| $D_f$ | [kg m s⁻³] | dissipation of free energy of flow into different kind of energy |
| $v$ | [m s⁻¹] | velocity of runoff, parallel to bed slope |
| $v_T$ | [m s⁻¹] | vertical fraction of v |
| $\rho$ | [kg m⁻³] | density of water with value of 1000 |



| | | |
|---|---|---|
| $g$ | [m s$^{-2}$] | gravitational acceleration with value of 9.81 |
| $Q$ | [m$^3$ s$^{-1}$] | discharge |
| $h$ | [m] | water height above hillslope end (h=z+d) |
| $B$ | [m] | hillslope width |
| $P_{eff}$ | [m s$^{-1}$] | effective rainfall intensity |
| $d$ | [m] | water column depth of surface runoff |
| $n$ | [m$^{-1/3}$ s] | manning coefficient |
| $S$ | [-] | slope of bed level |
| $z$ | [m$^2$ s$^2$] | geopotential of bed level to reference level |
| $X$ | [m] | length of hillslope, parallel to reference surface |
| $L$ | [m] | length of hillslope, parallel to bed level |
| $R$ | [m] | hydraulic radius |
| $A$ | [m$^2$] | wetted area of discharge |
| $r$ | [m] | radius of semi-circled rills |
| $m$ | [-] | number of semi-circled rills |
| $\tau_b$ | [kg m$^{-1}$ s$^{-2}$] | bed shear stress |
| $c_{FC}$ | [-] | Flow accumulation coefficient of CATFLOW-RILL model |
| $\alpha, \beta, \gamma$ | [radians] | Angles of CATFLOW-RILL hillslope surface |
| $Re$ | [-] | Reynolds number of surface runoff |
| $f_{crit}$ | [kg m$^{-1}$ s$^{-2}$] | Critical erosion force of surface runoff |

**2.2 The hillslope surface as open thermodynamic system**

To frame surface runoff processes into a thermodynamic perspective we define the surface of a hillslope as an open thermodynamic system (OTS; Kleidon, 2016). In this sense, the hillslope exchanges mass, momentum, energy and entropy with its environment (Fig. 1 and 2). Rainfall adds mass at a certain height and thus free energy in the form of potential energy
along the upper system boundary. Mass and free energy leave the system at the lower boundary due to surface runoff or via infiltration as subsurface flow (Zehe et al., 2013). In order to analyse the distribution of energy conversion processes in space we distinguish open thermodynamic subsystems ($OTS_{sub}$) that are assumed to be in steady state as shown in Fig. 2.





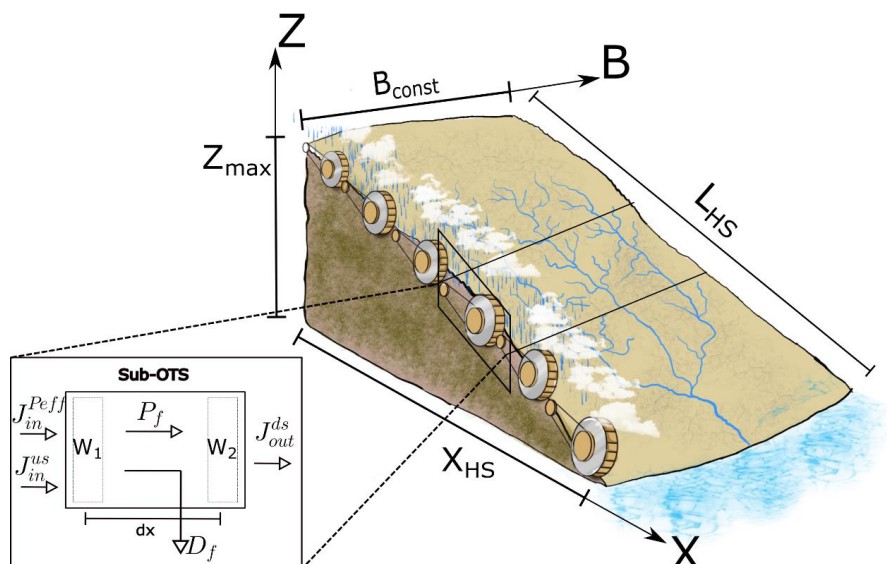

**Figure 2: Hillslope open thermodynamic system with spatial division into sub-OTS and energy flows from upstream and**

180 **downstream ($J_{in}^{us} = J_{f,in}^{pe} + J_{f,in}^{ke}$ and $J_{out}^{ds} = J_{f,out}^{pe} + J_{f,out}^{ke}$) as well as free energy reservoirs W₁ and W₂. Each control volume (sub-**

**OTS) is represented by a pony brake (cf. Leopold and Langbein, 1962)**

For each $OTS_{sub}$ we apply Eq. (4) in steady state and express potential and kinetic energy of the fluxes through hydraulic

variables (see Appendix A for derivation) to obtain:

$$
\begin{aligned}
D_f(x) &= J_{f,net}^{pe}(x) + J_{Peff,net}^{pe}(x) + J_{f,net}^{ke}(x) \\
&= \rho * g * \left( -\frac{dQ(x)}{dx} * h(x) - \frac{dh(x)}{dx} * Q(x) + P_{eff}(x) * h(x) * B(x) \right) \\
&\quad - \frac{1}{2} * \rho * \left( \frac{dQ(x)}{dx} * v(x)^2 + 2 * v(x) * \frac{dv(x)}{dx} * Q(x) \right)
\end{aligned}
\tag{5}
$$

Where Q [m³ s⁻¹] is the overland flow rate, v [m s⁻¹] the average flow velocity, h [m] the water height above the hillslope outlet,

ρ [kg m⁻³] the density of water, g [m s⁻²] gravitational acceleration and P_eff [m s⁻¹] the difference between rainfall intensity and

infiltration. The energy, which is dissipated per unit length depends on the net potential plus the net kinetic energy flow plus

the additional energy input per time through precipitation. With the assumption that change of velocity in space is close to zero

(v=v_const) and $\frac{dQ(x)}{dx} = P_{eff} * B(x)$ Eq. (5) becomes:

$$
D_f(x) = -P_{eff} * \rho * g * B(x) * \frac{v_{const}^2}{2 * g} - Q(x) * \rho * g * \frac{dh(x)}{dx}
\tag{5 a}
$$

For Eq. 5a, we can see that the first term scales with P_eff and the second with Q. With precipitation usually decreasing and

190 discharge increasing in downstream direction, there will be a flow path length where we can reduce Eq. (5 a) to its second

term:

$$
D_f(x) = -Q(x) * \rho * g * \frac{dh(x)}{dx}
\tag{5 b}
$$

This could be the case for a larger stream and explain how systems with small variation of mass minimize energy dissipation

by flattening of their geopotential gradients ($\frac{dh(x)}{dx}$ approaches very small values). As more and more mass accumulates along

the flow path, dissipation of discharge power is less controlled by changes in velocity or mass. This results in an increasingly





flattened geopotential gradient and therefore a negative exponential distribution of geopotential (Leopold and Langbein, 1962). In the literature Eq. (5b) is also called stream power (Bagnold, 1966) and is used to calculate the force $\tau$ [N m$^{-2}$] that acts on bed material per unit area ("shear stress", with $d$ [m], as depth of water column):

$$\tau(x) = \frac{D_f(x)}{v(x) * b(x)} = -d(x) * \rho * g * \frac{dh(x)}{dx} \tag{6}$$

### 2.3 Steady state spatially distributed energy of overland flow: Rivers vs. hillslopes

The steady state distribution of energy and its gradients within a system stands in feedback with the flow and flow accumulation. The larger the gradient, the higher the flow, which leads to a faster depletion of the gradient and less flow. In hydraulic sciences the geopotential gradient of a runoff system is usually approximated with the slope $S$ [m m$^{-1}$] of the riverbed (Bagnold, 1966), reducing Eq. (6) to the depth slope product and facilitating the calculation of depth averaged momentum balances by use of a general friction law:

$$Q = c_1 * b * d^{c_2} * \sqrt{S} \tag{7}$$

Where c1 and c2 are coefficients which vary for Manning-Strickler (Manning's n [s m-1/3]), Chezy (C [m1/3 s-1]) and Darcy-Weißbach (f [-]) (Singh et al., 2003).

**Table 2: Coefficients of general friction law**

|  | $c_1$ | $c_2$ |
|---|:---:|:---:|
| **Manning-Strickler** | $\frac{1}{n}$ | $\frac{5}{3}$ |
| **Chezy** | $C$ | $\frac{3}{2}$ |
| **Darcy-Weißbach** | $2 * \left(\frac{2g}{f}\right)^{0.5}$ | $\frac{3}{2}$ |

To highlight the difference between rivers and hillslopes Figure 3 compares the spatial distributions of potential energy along a flow path in a river system and on a hillslope. We take the Rhine from Basel to its mouth at Emmerich as an example, and we calculated potential energy $E_{pot}$ [kg m$^2$ s$^{-2}$] and volumetric potential energy $E_{pot}^{SP}$ [kg m$^{-1}$ s$^{-3}$] ($E_{pot}^{SP} = E_{pot}/Q$) from the mean yearly discharge and the water depth above sea level (data obtained from LUBW, 2021). Furthermore, we digitized the results of Emmett (1970) from his experiments on hillslope plots and computed $E_{pot}$ and $E_{pot}^{SP}$ from measured water depth above outlet reference level and mean flow velocity.



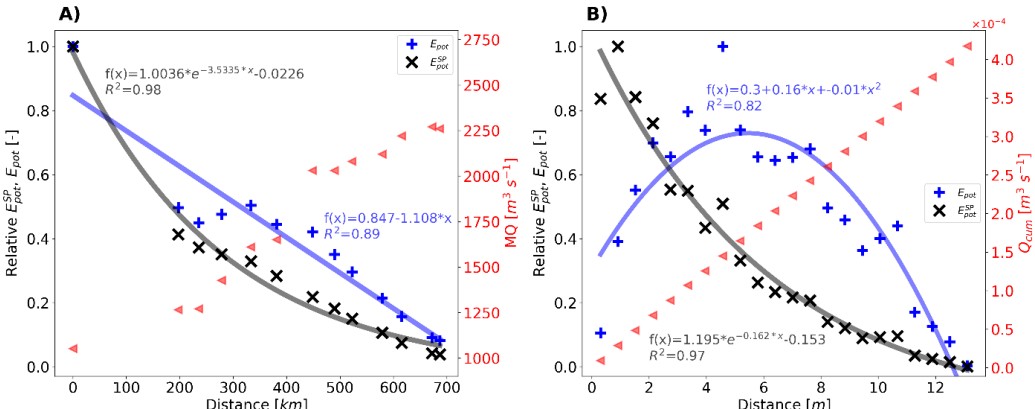

**Figure 3: a) Relative volumetric-, total potential energy and average annual discharge *MQ* along the River Rhine; b) Relative volumetric-, total potential energy and accumulated discharge *Q$_{cum}$* on hillslope plot (experiment "New Fork River Site 1", Emmett (1970))**


From the plots in Figure 3, we can see that in river and hillslope systems alike, volumetric energy $E_{pot}^{SP}$ as well as its gradient minimize along the flow path. Or differently stated, the energy expenditure per unit discharge minimizes in downstream

direction (solid black lines). This is very much in line with the principle of Rodriguez Iturbe et al. (1992) and Yang (1976). Interestingly however, the distribution of total potential energy along flow path is different for hillslope and river systems. The river Rhine distributes the loss of potential energy equally along its flow path and thus performs work uniformly on its course, as stated in the second principle of Rodriguez-Iturbe et al. (1992). The distribution of total energy on hillslopes follows a quadratic function, which rises to a maximum at 6 m and declines further downslope. This finding is on the one hand

straightforward, because if mass is accumulatively added along the flow path, whilst loosing geopotential, there must be a maximum somewhere downslope. On the other hand, we think this finding is astonishing as a positive energy gradient in downslope direction poses somewhat of a challenge to our common friction laws. In section 3 we explore how this maximum in potential energy depends on rainfall intensity and hillslope form.

We have already pointed out, that surface runoff on hillslopes has a transitional character. As also measured by Emmett (1970), the flow regime upslope of the hillslope starts laminar, transitioning to mixed and finally turbulent flow further downslope. Upslope, very shallow water depths limit erosion to the impact by raindrops, so called splash erosion and soil creep, which is erosion through gravity, while further downstream erosion by turbulent drag of the water flow is dominant (Shih and Yang, 2009). Similarly, there exists a minimum flow path length for surface runoff to occur, sometimes at the beginning referred to

thread flow, which transitions into shallow sheet flow (Dunne and Dietrich, 1980) and on bare soils even farther downslope forms rill flow. This transitional character of runoff on hillslopes is represented in Figure 4.





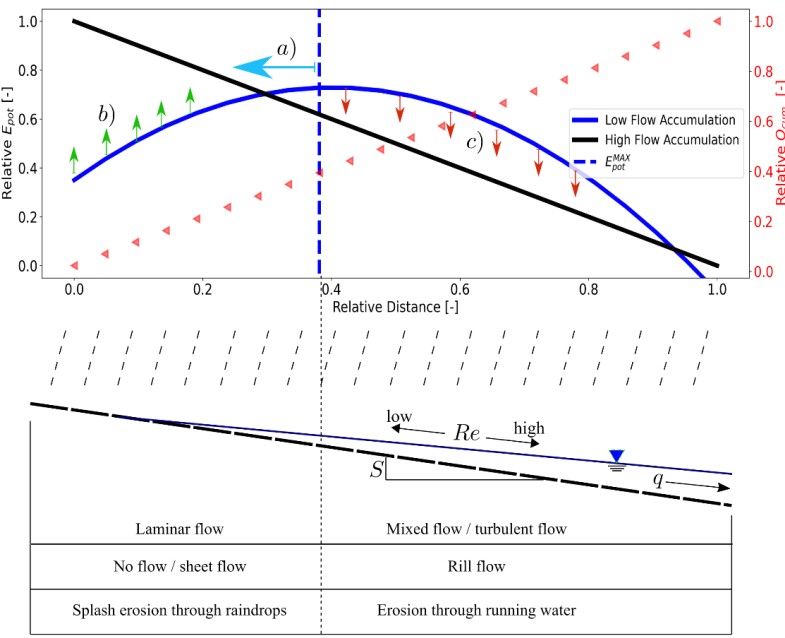

**Figure 4: Simplification of overland flow processes on hillslopes (modified after Shih and Yang (2009)) as a function of Reynolds number *Re* and distribution of potential energy**

In section 4, we explore whether this transition of flow regimes on hillslopes (from laminar to turbulent or sheet to rill flow) relates to the distribution of energy and its local maximum. Our rationale is that laminar flow is less dissipative than turbulent flow, and laminar conditions might therefore be related to the build-up of energy (Figure 4, relative distance 0 until $E_{pot}^{MAX}$), while the stronger dissipative character of mixed and turbulent flow should be related to a decrease of total energy (Figure 4, relative distance $E_{pot}^{MAX}$ until 1.0). We want to stress that we speak of laminar flow if there is a clear dependence between flow

Reynolds number of surface runoff and friction coefficient (Phelps, 1975). For purpose of comparison with earlier studies of hydraulics of surface runoff (Emmett, 1970; Parsons et al., 1990) we calculate flow Reynolds number *Re* as per Eq. (8), relating the characteristic length of Hortonian surface runoff to flow in a fully filled circular pipe. Here, *v* represents the depth averaged flow velocity, *R* the hydraulic radius and $v$ is the kinematic viscosity with a value of $10^{-6}$ [m$^2$ s$^{-1}$].

$$Re = 4 * \frac{v * R}{v} \tag{8}$$

## 3. A first order assessment of macro-topographic controls on Hortonian surface runoff and related energy conversions

**3.1 Definition of hillslope forms and width functions**

In this section, we explore how typical hillslope configurations and effective rainfall forcing, control runoff accumulation and related energy conversions. We distinguish four typical hillslope forms, characterized by either a linear, sinusoidal, exponential or a negative exponential geopotential function along a representative flow path x (Figure 5a):

$$z_{lin}(x) = -\frac{z_{max}}{x_{HS}} * x + z_{max} \tag{9 a}$$

$$z_{sin}(x) = \frac{z_{max}}{2} * \cos\left(\frac{x}{x_{HS}} * \pi\right) + \frac{z_{max}}{2} \tag{9 b}$$

$$z_{exp}(x) = e^{-x*2*k} * z_{lin}(x) \tag{9 c}$$

$$z_{neg}(x) = -e^{x*k} * -z_{lin}(x) \tag{9 d}$$





All hillslope forms start at $z_{max}$, the maximum specific geopotential in m$^2$ s$^{-2}$, and end at zero, depleting all available
geopotential gradients. In our examples, we assumed $z_{max}$ as the specific geopotential of 10 m altitude multiplied by the gravity
of the earth of 9.81 m$^2$ s$^{-2}$, and a projected hillslope length $X_{HS}$ of 100 m. $k$ is a smoothing factor for the exponential functions
and equals 0.01 m$^{-1}$. These forms have been chosen as they represent the different geomorphological stages of a hillslope under
erosion in time, starting with $z_{neg}$ as the youngest formation (largest gradients towards the end) and ending with $z_{exp}$ and $z_{sin}$ as
older formations (smaller gradients towards the end). We then combine these forms with three different width distributions,
which are either constant, converging or diverging (Figure 5a, and b). In our analysis we keep the projected area constant at
5000 m$^2$ for all configurations, which results in an equal total surface runoff from all hillslope forms for a given effective
rainfall intensity. Finally, we computed steady state surface runoff for effective rainfall intensities of 5-, 10-, 20- and 50-mm
hr$^{-1}$ either without runon (Q$_0$=0, Figure 5c) or with 20 kg s$^{-1}$ runon (Q$_0$=0.02 m$^3$ s$^{-1}$, Figure 5d), which is roughly a quarter of
the maximum accumulated runoff at 50 mm hr$^{-1}$ rainfall. It should be noted that we considered one case with no rainfall and
runon only (I=0; Q$_0$>0, Figure 5d). We included this scenario to highlight the differences between runon without rainfall
accumulation and runoff with rainfall accumulation when calculating spatial energy dynamics. The differently dotted lines in
Figure 5b, c, and d represent the three hillslope width distributions and show their influence on runoff accumulation.
Nevertheless, the total runoff at the end of the hillslope is independent of width distribution as the projected area remains equal
for all hillslope forms.


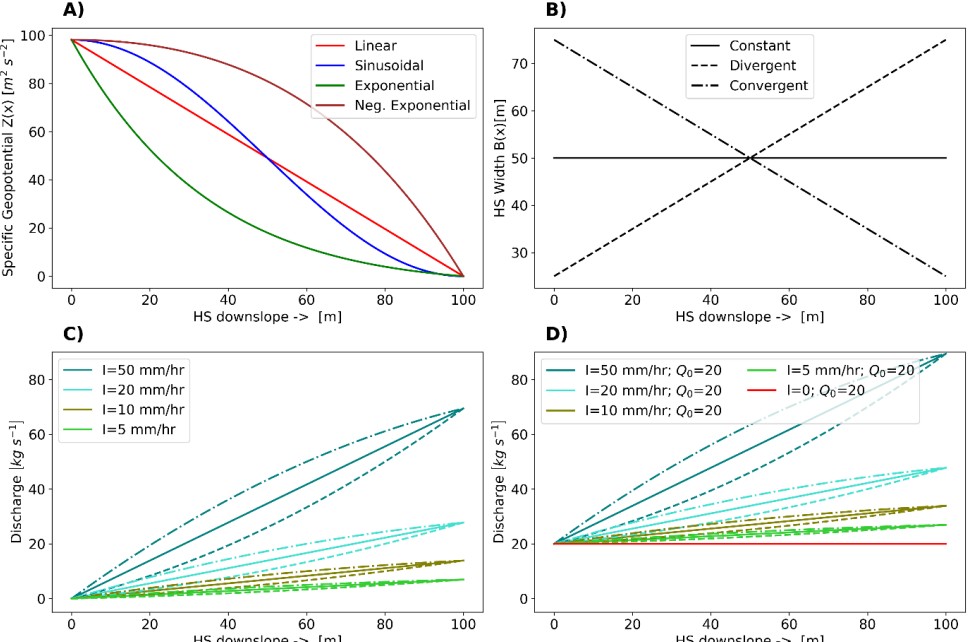

**Figure 5: Topography and width of the different hillslopes (panels a and b), resulting steady state discharge along the hillslope for
the case of no and constant runon of 20 kg s$^{-1}$ (panels c and d). The line types in panels c and d correspond to the width functions
in panel b**

For all combinations of runoff accumulation and hillslope topography, we computed the steady state spatial distribution of
water mass and flow velocity using Eq. (7) and Manning's $n$=0.1 s m$^{-1/3}$. From the spatial distribution of energy, we then
computed fluxes of potential flow energy $E_f^{pe}$, kinetic flow energy $E_f^{ke}$, effective rainfall and finally energy expenditure $D_f$
per unit flow length with Eq. (4) (see Appendix A for details of computation).





### 3.2 Spatial maxima of potential energy

Generally, we found that the trade-off of downslope mass accumulation and declining geopotential leads to a distinct potential energy maximum, which has a clear dependence on the slope form, width function and strength of rainfall forcing (Figure 6). This implies that the hillslope can be sub-divided into three classes of spatial energy dynamics:

1) $\frac{dE_f^{pe}(x)}{dx} > 0$

2) $\frac{dE_f^{pe}(x)}{dx} = 0$

3) $\frac{dE_f^{pe}(x)}{dx} < 0$

Within the first interval potential energy flux increases along the flow path, as the additional mass from rainfall adds more energy to the sub-OTS than flows out. At a certain distance (interval 2), energy outflow equals energy input through precipitation plus upstream inflow and we observe an energetic maximum. Within the third interval, energy outflow is continuously larger than energy inflow, effectively depleting the accumulated geopotential of interval 1.

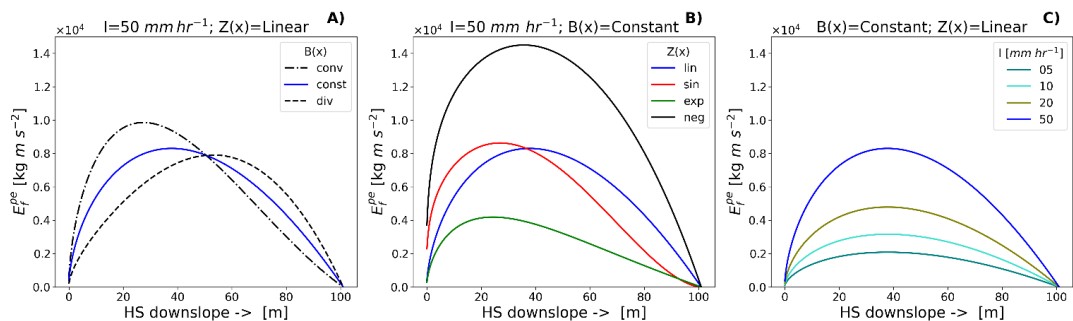


**Figure 6: Distribution of potential energy $E_f^{pe}$ per unit length in [Joule m⁻¹] as a function of a) hillslope width b) geopotential distribution (form) and c) rainfall intensity I**

Figure 6a shows that the location of the energetic maximum moves upslope when changing the width function from divergent (*div*), over parallel (*const*) to convergent (*conv*). The magnitude of the absolute value of the maximum increases in a similar

fashion. The distribution of geopotential from top to bottom clearly influences the location and size of maxima (Figure 6b). Interestingly a hillslope with a negative exponential form, which is morphologically the youngest, has by far the largest potential energy maximum and therefore highest geopotential difference with the hillslope end. Larger differences mean more available potential energy to perform work within the specified hillslope distance, which might result in enhanced erosion in comparison to e.g. sinusoidal or exponential hillslope forms. In line with this idea about morphological ages is also the growth

of energy gradients from exponential and sinusoidal to negative exponential (old, smaller gradients to young, larger gradients). Similarly, an increasing rainfall intensity increases the magnitude of the energy maxima while it does not affect their location (Figure 6c). Increasing energy maxima imply steeper energy gradients resulting in more power during the energy conversion processes. We thus state that the distribution of potential energy in space as a function of hillslope width, form and rainfall intensity seems to go hand in hand with the morphological stages of hillslope forms.

The analysis of hillslopes that accumulate rainfall and also experience a significant upslope runon (Figure 5d) revealed that the distribution of potential energy resembles much more a river's steady state with a continuously negative gradient (see Appendix B). The energetic dynamics of a hillslope with significant runon is therefore different to a hillslope with very little to no upslope runon.





### 3.3 Spatial patterns of stream power

In a second step, we calculated stream power $D_f$ (Eq. 5) per unit flow length in watt per meter (Figure 7) as well as per unit area $D_{f,A}$ ($D_{f,A} = D_f/B$) in watt per square meter (Figure 8) and plotted the resulting spatial distributions to the climatic-topographic combinations as shown in Figure 5a, b, and c (no runon is considered here).

For all hillslope forms, our calculations show that compared to a constant width, a converging width function increases $D_f$ and a diverging width function decreases $D_f$ (Figure 7a). If more rainfall falls at higher geopotentials (converging widths), the

available potential energy to be converted is larger than on hillslopes with diverging widths, which accumulate the larger share of runoff at the lower end. We also note in Figure 7a that a converging width results in a limitation of the growth of power per unit length in contrast to diverging and constant widths, which increase power due to the additional mass input in downslope direction. Figure 7b shows how the geopotential distribution $z(x)$ influences $D_f$. Exponential and sinusoidal distributions result in a point along the flow path where energy conversion is maximized, whereas negative exponential and linear distributions

unlimitedly increase power in downslope direction. For hillslope forms with a power limitation (sinusoidal, exponential) converging widths lead to a power maximum that is relatively farther upstream than it is the case for diverging widths. Figure 7c reveals that the rainfall intensity merely has a linear scale effect on the magnitude of $D_f$ and does not influence its relative spatial distribution.

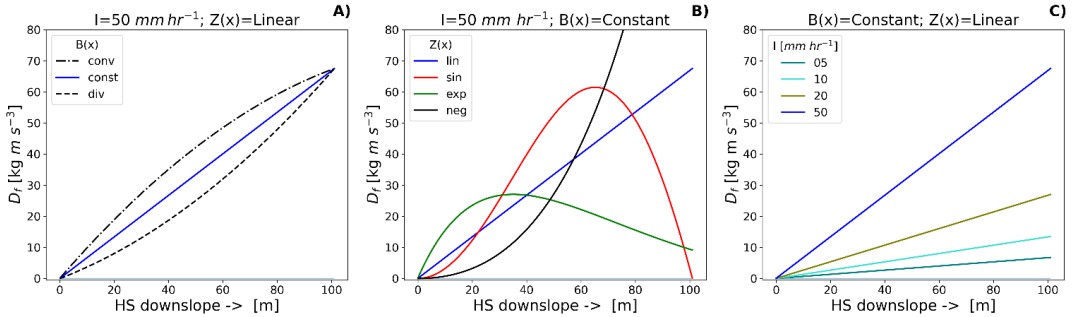

**Figure 7: Spatial distribution of stream power $D_f$ per unit flow length [W m⁻¹]; a) Rainfall intensity 50 mm hr⁻¹ and linear hillslope form but varying hillslope widths (constant, converging, diverging); b) Rainfall intensity 50 mm hr⁻¹ and constant hillslope width but varying hillslope forms (linear, sinusoidal, exponential, negative exponential); Linear hillslope form and constant hillslope width but varying rainfall intensities;**

While stream power per unit length represents macroscopic energy dynamics of the whole hillslope and therefore total energy,

stream power per unit area is related to forces that act locally on soil material. Figure 8a shows the distribution of unit stream power $D_{f,A}$ (in watt per square meter) for the different width functions (compare Figure 5b) of a hillslope with linear distribution of geopotential. In contrast to a constant width, diverging widths decrease stream power per unit area and converging width increase $D_{f,A}$ in downslope direction. Although the additional mass per unit area for converging widths decreases in downslope direction, flow accumulation leads to larger water depths and therefore more power per unit area. It should be noted, that at

the hillslope outlet, total stream power (Figure 7a) is equal for all hillslope width functions, while stream power per unit area is clearly different (Figure 8a). $D_{f,A}$ of the considered geopotential distributions is shown in Figure 8b, mirroring a scaled distribution of total stream power (Figure 7b), which indicates that while the distributions of $D_f$ and $D_{f,A}$ are controlled by energy gradients, the magnitude of macroscopic stream power is controlled by hillslope width functions and therefore flow accumulation. A hillslope which is accumulating more flow along the same distance has larger stream power than a hillslope

with less flow accumulation, however both hillslopes might in total dissipate the same amount of energy for the same flow length. Similarly, rainfall intensity has just a scale effect on $D_{f,A}$ but does not influence its relative spatial distribution (Figure 8c).





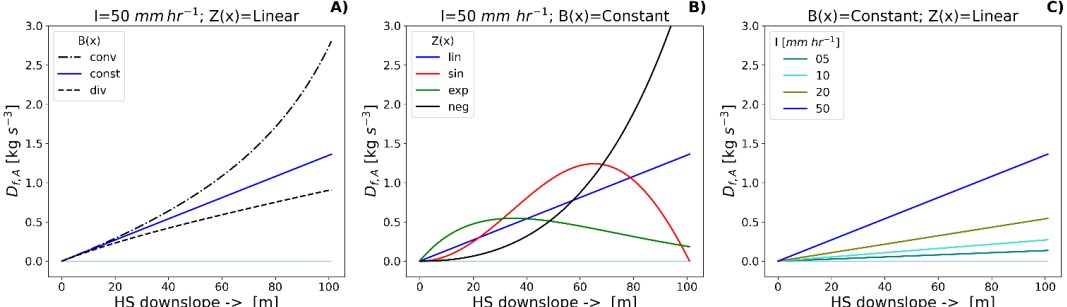

**Figure 8: Spatial distribution of stream power D$_{f,A}$ per unit flow area [W m$^{-2}$]; a) Rainfall intensity 50 mm hr$^{-1}$ and linear hillslope**
**form but varying hillslope widths (constant, converging, diverging); b) Rainfall intensity 50 mm hr$^{-1}$ and constant hillslope width**
**but varying hillslope forms (linear, sinusoidal, exponential, negative exponential); Linear hillslope form and constant hillslope width**
**but varying rainfall intensities;**

**4 Numerical simulation of overland flow experiments and their micro-topographic controls on distributed energy**
**dynamics**

We now explore the spatial distribution of potential energy in sheet and rill overland flow within rainfall-runoff experiments

carried out in the Weiherbach catchment (Gerlinger, 1996). Therefore, we build an extension to the physical hydrological

model Catflow, which allows the accumulation of flow from sheet flow areas into rills (Catflow-Rill).

**4.1 Study area and experimental data base**

The Weiherbach catchment is an intensively cultivated catchment which is almost completely covered with loess up to a depth

of 15 m (Scherer et al., 2012). It is located in the Kraichgau region northwest of Karlsruhe in Germany. Because of the hilly

landscape, the intensive agricultural use and the highly erodible loess soils, erosion is a serious environmental problem in the

Kraichgau region. The Weiherbach itself has a catchment area of 6.3 km$^2$ and is around 4 km long. Elevation ranges from 142

m to 243 m above sea level; the slopes are long and gentle in the west, and short and steep in the eastern part of the catchment.

The climate is semi-humid with a mean annual temperature of 10 °C (Scherer, 2008). More than 90 % of the catchment area

is arable land or pasture, 7 % are forested and 2.5 % are paved (farmyards and roads). Severe runoff and erosion events are

typically caused by thunderstorms in late spring and summer, when Hortonian overland flow dominates event runoff generation

(Zehe et al., 2001). A comprehensive hydro-meteorological dataset as well as data on soil hydraulic properties, soil erosion,

tracer and sediment transport are available for the Weiherbach (Scherer et al., 2012; Schierholz et al., 2000).

Here we use 2 selected characteristic rainfall simulation experiments (Gerlinger, 1996), which were performed to explore

formation of overland flow and the erodibility of the loess soils (Scherer et al., 2012). The rainfall simulators were designed

to ensure both realistic rainfall intensities and kinetic energies on plots of 2 m by 12 m size. Rainfall intensity of both

experiments was set to 62.4 mm h$^{-1}$. Rainfall simulation was stopped when overland flow and sediment concentration had

reached steady state. Runoff and sediment concentrations in overland flow samples were derived from samples taken during

the experiments. The different sites were characterized according to their antecedent soil moisture, soil texture and organic

content in the upper 5-10 cm (Scherer et al., 2012). Additionally, surface roughness (Manning's n) was estimated from the

falling limb of the observed hydrograph (Engman, 1986; Govers et al., 2000). For the two selected experiments (lek_2 and

oek2_4), antecedent soil moisture of the plots was 25.6 and 18.4 Vol%, organic content 1.9 and 2.5 %, clay content 16.8 and

21.1 % and Manning's $n$ 0.045 and 0.032 s m$^{-1/3}$. Further details on the experimental setup are provided by Gerlinger (1996),

Seibert et al. (2011), and Scherer et al. (2012).



### 4.2 Model and model setup

In order to analyse total spatial distribution of energy and dissipation we present here an extension to the Catflow model (Zehe et al., 2001), accounting for a dynamic link between sheet- and rill flow of Hortonian surface runoff. The model has previously been extended, incorporating water-driven erosion (Scherer, 2007) and has been shown to successfully portray the interplay of overland flow, preferential flow and soil moisture dynamics from the plot to small catchment scales (Graeff et al., 2009; Loritz et al., 2017; Zehe et al., 2005, 2013).

A catchment is represented in CATFLOW by a set of two-dimensional hillslopes (length and depth), which may be connected by a river network. Each hillslope is discretized using curvilinear orthogonal coordinates; the third dimension is represented by a variable width. Subsurface water dynamics are described by Richards' equation, which is solved numerically by an implicit mass-conservative Picard iteration scheme. The simulation time step for soil water dynamics is dynamically adjusted to achieve an optimal change of the simulated soil moisture, which assures fast convergence of the Picard iteration. Soil hydraulic properties are usually parameterized using the van Genuchten-Mualem model (Mualem, 1976; van Genuchten, 1980), but other options are available. Enhanced infiltrability due to activated macropore flow is conceptualized through enlarging the soil hydraulic conductivity by a macroporosity factor $f_{mak}$, when a soil moisture threshold is exceeded. This approach is motivated by the experimental findings of Zehe and Flühler (2001a and 2001b) in the Weiherbach catchment and has been shown to be well suited for predicting rainfall-runoff dynamics (Zehe et al., 2005) as well as tracer transport at the plot and the hillslope scales (Zehe and Blöschl, 2004; Zehe et al., 2001).

### 4.2.1 Representation of overland flow in Catflow and Catflow-Rill

Overland flow is simulated in Catflow-Rill with the diffusion wave equation, which is numerically solved using an explicit upstreaming scheme, a simplification of the Saint-Venant equations for shallow water flow, for details of the numerical scheme we refer to Scherer (2007). Flow velocity is calculated with Manning's equation (Eq. 7). The previous Catflow model assumes sheet flow only. To incorporate a rill domain that dynamically interacts with sheet flow, we conceptualise the hillslope surface similar to the open book catchment (Wooding, 1965) as an open book hillslope (Figure 9). In this configuration water may accumulate in a trapezoidal rill of width $B_r$ in the middle of the open book hillslope with width $B_{HS}$ and downslope length $L_{HS}$. Rainfall is added proportionally to the projected area along the flow path in both domains, resulting in spatially distributed sheet flow $Q_{SF}$ and rill flow $Q_{RF}$. The link is established by a flow accumulation coefficient $c_{FC}$ (Eq. 12). This is visualized in Figure 9 by the angle $\gamma$ (in radians) between the vectors $\overrightarrow{Q_{SF}}$ and $\overrightarrow{Q_{RF}}$, which manifest at each point on the sheet flow surface the tendency of a volume water to flow downslope the hillslope gradient $\alpha$ or to follow the secondary flow accumulation gradient $\beta$ (Eq. 12).

$$dQ_{link}(x) = Q_{SF}(x) * c_{FC}(x) \tag{12}$$

$$tan(\gamma) = \frac{|\overrightarrow{Q_{RF}}|}{|\overrightarrow{Q_{SF}}|} = \frac{\alpha}{\beta} \tag{13}$$

The maximum amount of flow which is transferred per unit flow path length from the sheet domain into the rill domain is then:

$$c_{FC,max} = \gamma * \frac{2}{\pi} \tag{14}$$

However, depending on the configuration of the open book hillslope, we need to account for a flow path length $L_{FC}$, where flow accumulation becomes constant and maximum:

$$L_{FC} = B_{HS} * tan(\gamma) \tag{15}$$

From hillslope top to the flow path length $L_{FC}$, the flow accumulation coefficient is linearly interpolated between $c_{FC}(x = 0) = 0$ until $c_{FC}(x = L_{FC}) = c_{FC,max}$.





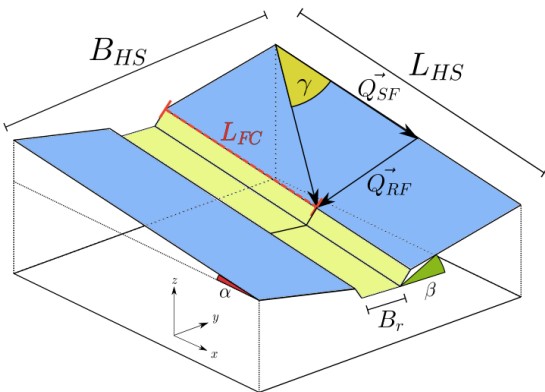

**Figure 9: Representation of overland flow domains in Catflow-Rill as open book hillslope: Sheet flow domain (blue area) and Rill flow domain (yellow area).**

**4.2.2 Model setup and calibration of flow accumulation**

From the experimental database Scherer et al. (2012) created Catflow simulation setups, which were calibrated to reproduce runoff by adapting the macroporosity factor to scale infiltration capacity. The hillslopes were parameterized and initialized

using observed data on average topographic gradient, plant cover, soil hydraulic functions, surface roughness, soil texture, and antecedent soil moisture. The models were driven by a block rain of the respective intensity and duration of the experiment. For calibration with the here presented extension Catflow-Rill (see previous section), we chose 2 characteristic experiments with similar slope and equal rainfall intensities of 62.4 mm h$^{-1}$. For experiment "lek_2" (slope=0.163 m m$^{-1}$) significant rill flow was reported (Gerlinger, 1996) with steady state rill runoff velocities ($v_{RF,obs} = 0.239\ m\ s^{-1}$) almost double the average

sheet flow velocities ($v_{SF,obs} = 0.122\ m\ s^{-1}$). The reported rill velocities were measured by upslope tracer injections and correspond to the time it took until the peak of tracer concentration reached the plot outlet. Contrarily, during experiment "oek2_4" (slope=0.151 m m$^{-1}$) little to no rill flow was observed, manifesting in almost equal surface runoff velocities of $v_{SF,obs} = 0.142\ m\ s^{-1}$ and $v_{RF,obs} = 0.15\ m\ s^{-1}$. From here onwards subscript "*obs*" relates to measured, reported values from the referenced experimental studies and subscript "*sim*" relates to the results of the presented calibrated numerical

simulations. Both hillslopes were discretized on a 2D numerical grid with an average lateral distance of 60cm and vertically increasing distances starting with 1cm at the surface and ending with 5cm on the soil bottom. This resulted in 21 x 29 computation points for both 12m long, 2m wide and 1m deep hillslope plots. Manning's *n* was determined during the experiments (Gerlinger, 1996) as 0.045 s m$^{-1/3}$ for "lek_2" and 0.032 s m$^{-1/3}$ for "oek2_4". Soil hydraulic parameters of the Van Genuchten-Mualem model were reported by Schäfer (1999), who conducted a soil hydraulic parameter campaign within the

Weiherbach catchment and classified five homogeneous soil types. From these, parameters from the C-horizon of Pararendzina soil type were used for the presented simulations (Scherer, 2007) in accordance to the location of the experimental plots within the catchment (see Table 3). Grain size distributions are available for plot experiment "lek_2", which consist of 15% clay, 78% silt and 7% sand, with mean particle diameter $d_{sed}$ between 20 to 30 μm (Scherer, 2007).

**Table 3: Soil hydraulic parameters of Van Genuchten-Mualem model for simulated hillslopes, namely saturated hydraulic conductivity $k_s$, saturated soil moisture $\theta_s$, residual soil moisture $\theta_r$, reciprocal air entry point $\alpha_s$, as well as soil hydraulic form parameters $n_s$ and $\gamma_s$**

| | $k_s\ [m\ s^{-1}]$ | $\theta_s\ [m^3\ m^{-3}]$ | $\theta_r\ [m^3\ m^{-3}]$ | $\alpha_s\ [m^{-1}]$ | $n_s\ [-]$ | $\gamma_s\ [-]$ |
|---|---|---|---|---|---|---|
| **Paradenzina** | 6.803*10$^{-7}$ | 0.444 | 0.066 | 0.51 | 2.24 | 0.71 |





To calibrate the observed flow velocities, we adjusted the flow accumulation coefficient $c_{FC}$, starting at 0.001 and incrementing in 0.001 steps. For each calibration run we compared the steady state values of $v_{RF,sim}$ and $v_{RF,obs}$ and stopped the

incrementation of $c_{FC}$ when the residual of both values was below 1% of $v_{RF,obs}$.

### 4.3 Simulation results

### 4.3.1 Catflow-rill results

For both hillslopes under consideration the calibration produced good results after few steps of incrementing the flow accumulation coefficient. For "lek_2" this resulted in $c_{FC} = 0.018$ and for "oek2_4" in $c_{FC} = 0.0032$ (Figure 10a and c). Total

mass is conserved as total simulated discharge $Q_{tot,sim}$ ( $Q_{tot} = Q_{RF} + Q_{SF}$ ) stays constant independent of $c_{FC}$ for all simulations, equalling the observed discharges (Scherer et al., 2012). Except for the onset of surface runoff, $Q_{tot,sim}$ stays with 10% error tolerance bands of measured total discharge $Q_{tot,obs}$ for both experiments (compare Figure 10a and c grey bands). The here presented Catflow-Rill simulations divide total surface runoff into simulated sheet flow $Q_{SF,sim}$ and rill flow $Q_{RF,sim}$, computing higher discharges in the rill domain with increasing flow accumulation coefficient. For lek_2, the final rill flow

velocity resulted in 0.238 m s$^{-1}$ and for oek2_4 in 0.15 m s$^{-1}$, matching the observed values 0.239 m s$^{-1}$ for lek_2 and 0.15 m s$^{-1}$ for oek2_4 (Figure 10b and c). Computed sheet flow velocities are close to observed steady state results but not as a precise match as calibrated rill flow velocity. One reason might be the measurement approach of sheet flow, which was done by indirect calculation of $v_{SF}$ through measured total discharge and $v_{RF}$ (Gerlinger, 1996), and therefore is likely to produce larger measurement errors. The final simulated steady state value of $v_{SF}$ is however for both experiments within a 10% error margin,

which is tolerable in the light of measurement uncertainty.



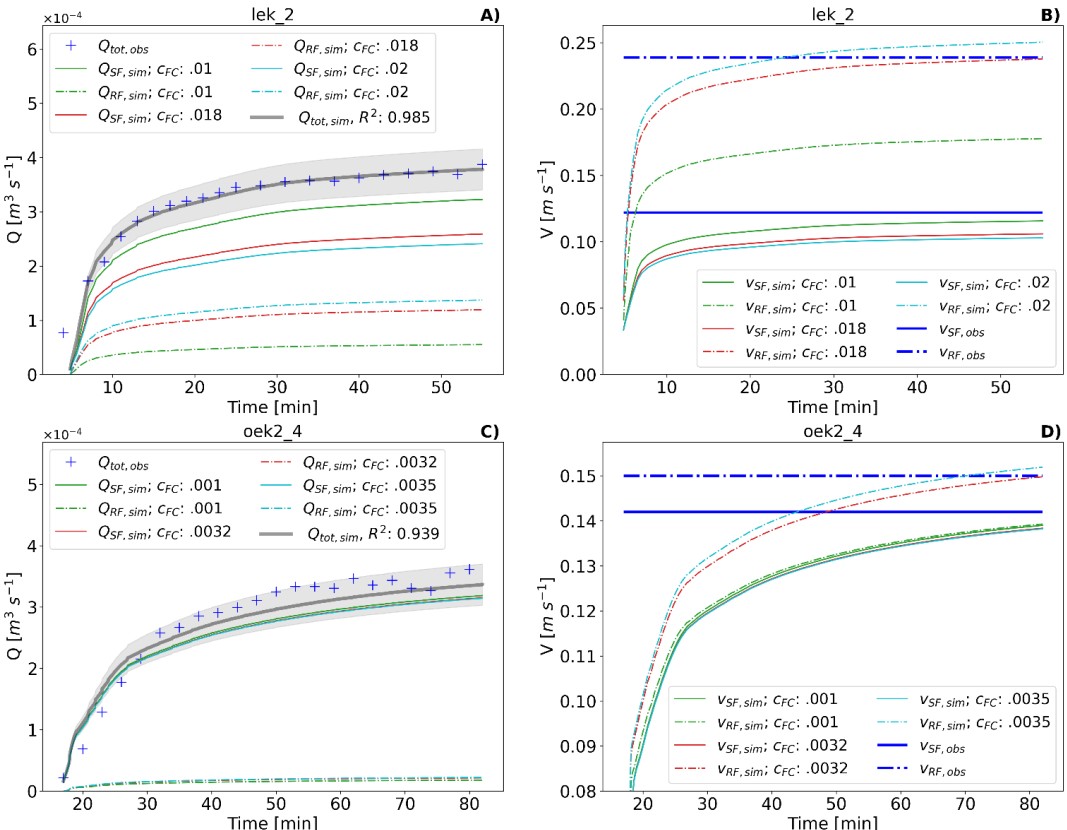

**Figure 10: Results of calibrations runs for both experiments: a) and c) Calibrated total discharge $Q_{sim,tot}$, measured discharge $Q_{tot,obs}$ (incl. grey 10% error band) and computed contributions of sheet flow $Q_{SF,sim}$ and rill flow $Q_{RF,sim}$; b) and d) Observed rill and sheet-flow velocities $v_{RF,obs}$ and $v_{SF,obs}$ and calibration runs for different flow accumulation coefficient $c_{FC}$**

### 4.3.2 Distribution of energy

The calibrated CATFLOW-Rill models provide an estimate of the spatial distribution of power and energy for the rill- and the sheet- domains. based on $Q_{SF}$ and $Q_{RF}$ (Figure 11a and c) and therefore allow a comparison of spatial energy distribution of systems with high accumulation of runoff in rills and systems with little to no rill flow. **Fehler! Verweisquelle konnte nicht gefunden werden.**a and Figure 12a show the spatial distribution of potential energy per flow length $E_f$ [J m⁻¹] for each domain and experiment. First of all, we note that both approaches of runoff calculation (Catflow and Catflow-Rill) result in a local maximum of potential energy and that more energy is stored within the sheet- than in the rill flow domain. The rill simulations increase potential energy within the rill domain and decrease $E_{pot}$ in the sheet flow domain. This happens non-linearly, meaning relatively more energy is transferred from the sheet to the rill flow domain downslope than upslope. As a result, the location of maximum potential energy is shifted in upslope direction. As in experiment lek_2 rill flow is much more pronounced, this shift and the related change of potential energy is much stronger than for oek2_4. Spatial distribution of kinetic energy $E_{kin}$ per flow length is plotted in Figure 11b and Figure 12b. While results of Catflow and Catflow-Rill are similar for oek2_4 (Figure 12b), the accumulation of flow in the rill domain has large impacts for lek_2 (Figure 11b). In comparison to the simulation without rill domain, kinetic energy decreases downslope, while in the rill flow domain $E_{kin}$ increases downslope. Interestingly, for Catflow-Rill, the kinetic energy at the hillslope outlet of sheet flow and rill flow are almost equal. The sum of both energies ($E_{pot} + E_{kin}$) is plotted as total free energy $E_{tot}$ per flow length in Figure 11c and Figure 12c. As potential energy is up to 1000 times larger in magnitude than kinetic energy, $E_{tot}$ is essentially equivalent to





$E_{pot}$. The plots show that the accumulation of flow in a rill reduces the amount of energy being stored as potential energy on the hillslope. Also, the spatial location of the local energy maximum shifts upslope.

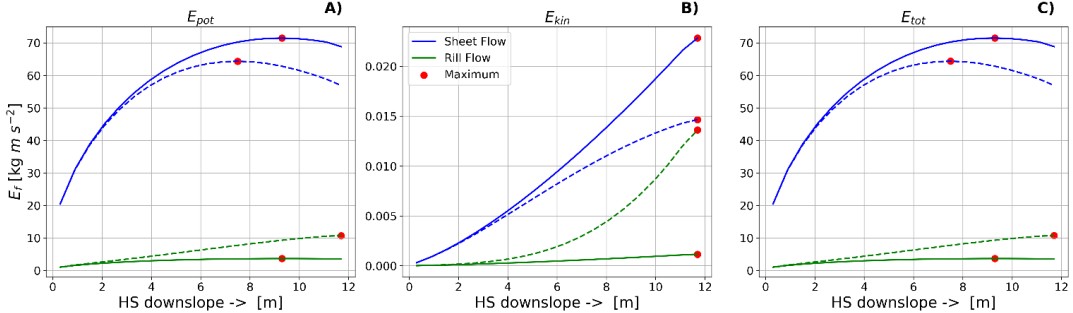

**Figure 11: Results of Catflow-Rill for no flow accumulation ($c_{FC} = 0$; solid lines) and flow accumulation ($c_{FC} = 0.018$; dashed lines) for experiment lek_2: a) Spatial distribution of potential energy $E_{pot}$ ; b) Spatial distribution of kinetic energy $E_{kin}$; c) Spatial distribution of total energy $E_{tot} = E_{pot} + E_{kin}$ of individual flow domains**

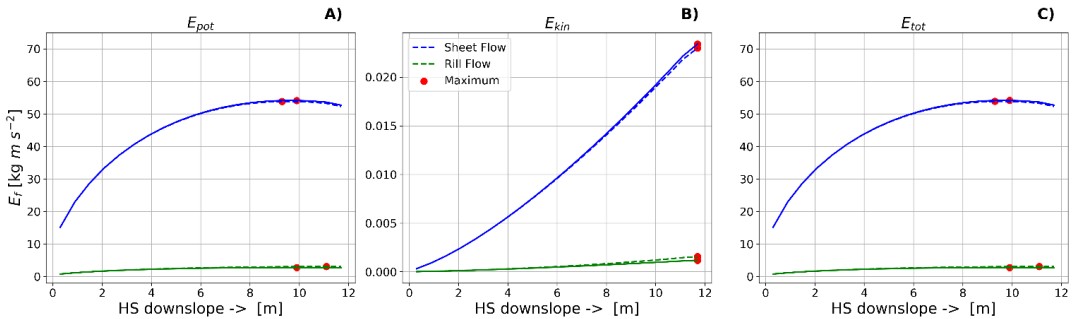

**Figure 12: Results of Catflow-Rill for no flow accumulation ($c_{FC} = 0$; solid lines) and with flow accumulation ($c_{FC} = 0.0032$;**
**dashed lines) for experiment oek2_4 2: a) Spatial distribution of potential energy $E_{pot}$ ; b) Spatial distribution of kinetic energy $E_{kin}$; c) Spatial distribution of total energy $E_{tot} = E_{pot} + E_{kin}$ of individual flow domains**

#### 4.3.3 Power and erosive force

From the Catflow and Catflow-Rill simulations we computed stream power per unit length $D_f$ [W m⁻¹] (Eq. 5) and per unit area $D_{f,A}$ [W m⁻²] ($D_{f,A} = D_f/b$) as well as bed stress $\tau$ [N m⁻²] (Eq. 6). Rill flow accumulation leads to larger $D_f$ and $D_{f,A}$
values within the rill domain and decreases both within the sheet flow domain (Figure 13a and b; Figure 14a and b). This result is much more pronounced for simulation of lek_2 due to the stronger rill flow component than for oek2_4. Simulation of oek2_4 shows little to no difference in power per unit length between the implementation with and without rill (Figure 14a), yet power per unit area is clearly larger in the rill- than in the sheet flow domain (Figure 14b). Similar results are found for bed stresses (Figure 14c and Figure 14c). Computation of surface runoff accumulation in a rill leads to larger forces per unit
area within the rill- and a lower $\tau$ within the sheet flow domain. For the soils of the experiments lek_2 and oek2_4 erosion resistance factors f_crit of 1.636 N m⁻² (lek_2) and 0.826 N m⁻² (oek2_4) were measured (Gerlinger, 1996), which we plotted as horizontal line in Figure 13c and Figure 14c. Interestingly, the effect on bed stress of rill flow in comparison to sheet flow for lek_2 becomes significant at the flow length where $\tau$ exceeds f_crit (Figure 13c).





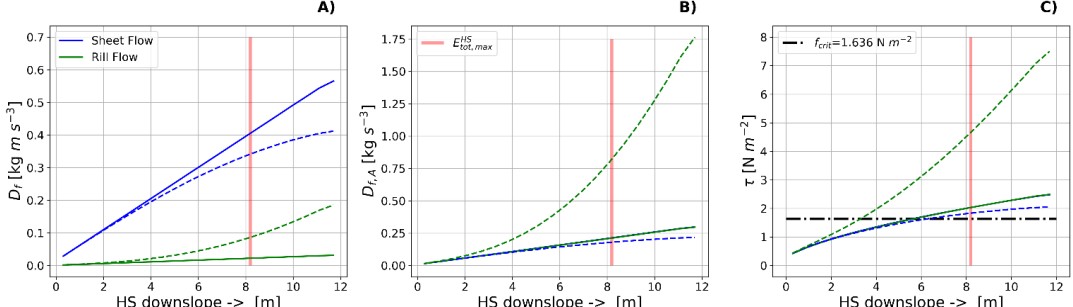

**Figure 13: Results of CATFLOW-RILL for no flow accumulation ($c_{FC} = 0$; solid lines) and flow accumulation ($c_{FC} = 0.018$; dashed lines) for experiment lek2: a) Spatial distribution of stream power per unit length $D_f$ [W m⁻¹]; b) Spatial distribution stream power per unit area $D_{f,A}$ [W m⁻²]; c) Spatial distribution of erosion force $\tau$**

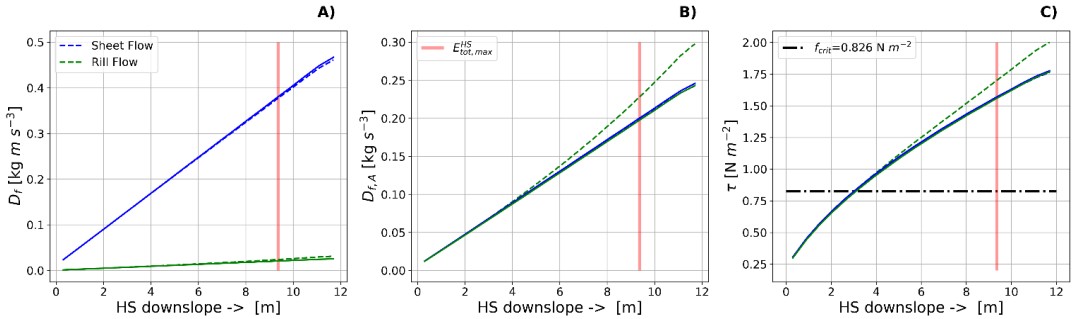

**Figure 14: Results of CATFLOW-RILL for no flow accumulation ($c_{FC} = 0$; solid lines) and flow accumulation ($c_{FC} = 0.018$; dashed**
**lines) for experiment oek2_4: a) Spatial distribution of stream power per unit length $D_f$ [W m⁻¹]; b) Spatial distribution stream power per unit area $D_{f,A}$ [W m⁻²]; c) Spatial distribution of erosion force $\tau$**

### 4.3.4 Energetic maxima and turbulent flow

As energy is additive, we can sum the potential and kinetic energy of rill- and sheet domain and compute the total free energy $E_{tot}^{HS} = E_{tot}^{SF} + E_{tot}^{RF}$ which is stored on the hillslope during steady state conditions (Figure 15a). Like for the individual results
of sheet- and rill flow (Figure 13 and Figure 14), total energy principally consists of potential energy. For the simulations of both experiments we observe a local maximum of $E_{tot}$ in space, which manifests for oek2_4 farther downslope than for lek_2. The total amount of energy stored on the hillslope is larger for lek_2 than for oek2_4, mostly due to the increased water flow depth as a result of a larger friction coefficient. For oek2_4 the computation of surface runoff with Catflow-Rill has little effect as little to no significant rill flow has been observed during the experiment. For simulation of experiment lek_2, Catflow-Rill
leads to less total energy which is stored on the hillslope and a shift of the local energy maximum in upslope direction.

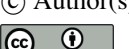



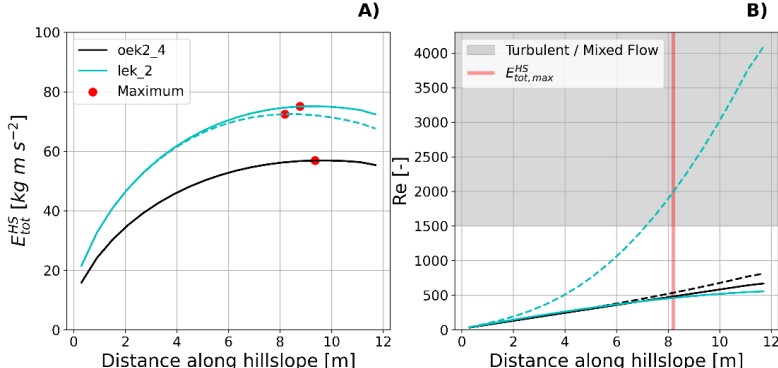

**Figure 15: Results of CATFLOW (solid lines) and CATFLOW-RILL (dashed lines) for simulations of experiments oek2_4 (S=0.151 m m⁻¹) and lek_2 (S=0.163 m m⁻¹): a) Distribution of total free energy $E_{tot}^{HS} = E_{pot}^{HS} + E_{kin}^{HS}$ per unit flow length; b) Distribution of Reynolds number Re (Eq. 8)**

For the analysis of flow regime, we plotted in Figure 15b flow Reynolds numbers (Eq. 8) of each simulation as function of hillslope length. According to Emmett (1970), surface runoff is considered of laminar flow regime up to Reynolds numbers of 1500. Between 1500 and approximately 6000, the flow regime switches from laminar to mixed and turbulent flow. For simulations with sheet flow only, maximum flow Reynolds numbers stay below 1000 for the entire hillslope. The only simulation where flow Reynolds number exceeds the critical value of 1500 (grey shaded area) and increases up to 4000 is the one with rill domain for the experiment lek_2. Note that flow Reynolds number increases exponentially up to the location of the total energy maximum (red vertical line, Figure 15b), and from there it grows linearly to hillslope toe.

## 5 Discussion

In this study we explore how potential energy and stream power of Hortonian surface runoff is controlled by macro- and micro-topography. To this end we establish a link between these controls and a thermodynamic perspective on conservation of energy and dissipation of free energy. As stream power is a flux of kinetic energy and therefore driven by energy gradients, we focus on the principal controls at the hillslopes scale namely the decline in geopotential, due to topography and the redistribution of water mass through flow accumulation. The analysis of typical hillslope forms, width functions and effective rainfall intensities revealed that the trade-off between mass accumulation and declining geopotential implies that potential energy of surface runoff does first increase downslope up to a maximum value and from there it declines. The location of this peak in potential energy of surface runoff changes with geopotential profile and width function of the hillslope. Interestingly, the most common hillslope forms in nature are those where the peak of energy is farther upslope (exponential distribution of geopotential, converging width) compared to hillslopes with negative exponential distribution of geopotential and diverging width. This suggests that hillslopes morphologically evolve in such a way that their distribution of energy in overland flow approaches more and more the configuration of a river system with a uniform energy expenditure along the flow path (Figure 4; cf. second principle in Rodriguez-Iturbe et al., 1992). We included this evolution process in Figure 4 by indicating the directions (small arrows) the distribution of $E_{pot}$ would evolve to. Flow accumulation leads upslope to shorter flow path lengths with increase of $E_{pot}$, shifting the point of energy maximum upslope (blue arrow). Downslope of the energy maximum, gradients flatten (red arrows) whilst upslope energy accumulates (green arrows). We propose that this evolution process is controlled by flow accumulation, in the form of macro- and micro-topography and to the extent that flow regime is a direct result of these controls. To further corroborate this idea, we would need to analyse detailed data on morphological development of a hillslope and surface runoff, which was beyond the scope of this study.





Furthermore, our results show that potential energy maxima vanish in case of an increasing upslope runon, and the hillslope

energetically resembles runoff behaviour of a river, which constantly expends energy. This is in line with what is observed in nature, as surface runoff increases, there will be a flow path length where topsoil is completely eroded and water flow is transported in a channel with a turbulent flow regime, marking a transitional behaviour of energy build-up on the hillslope, which is then used (dissipated) to transport material first within the rill system and then the stream channel. With increasing amount of runoff these channels extend farther upslope, effectively shortening flow path lengths of sheet flow (cf. Horton,

1945). The distribution of energy and its gradients along the flow path are the fuel of surface runoff and therefore stream power. Our results show that the distribution of geopotential gradients controls the distribution of stream power per unit length. Largest rates of energy expenditure are bound to highest differences in geopotential as only a small fraction of available potential energy is not dissipated and conserved as free energy (Loritz et al., 2019; Bagnold, 1966). In contrast, magnitude of stream power per unit area is controlled by flow accumulation of surface runoff. If the same amount of discharge flows down

two hillslopes with equal slopes but different widths, stream power per unit area is larger for the hillslope with smaller width, also increasing the force that acts on bed material. Erosion and therefore the redistribution of geopotential is a direct result of flow accumulation, a factor which is often not accounted for in macroscopic erosion models (Scherer, 2007). There are few studies which attributed downslope hydraulic geometric relations to flow accumulation (Parsons et al., 1990), highlighting that concentration of flow not only leads to more discharge per unit area and therefore stream power but also less resistance to

flow, effectively minimizing frictional losses.

On a micro-topographic level, we investigated the interplay between sheet- and rill flow for overland flow energy using data from rainfall-runoff simulation experiments. Therefore, we present a straightforward and successful numeric approach to separate surface runoff into sheet- and rill flow. Here we explored (i) whether maxima of potential energy do indeed emerge

on the short 12m experimental plots, (ii) how rills control their spatial location and magnitude, and (iii) whether these maxima are related to reported transitions of flow regime from laminar to turbulent flow on hillslopes, using our thermodynamic framework.

We found that observed total runoff and distinctions between rill and sheet flow velocities were well captured by calibrating the macroporosity factor to the former and the flow accumulation coefficient to match rill flow. We stress that these simulations

reproduce velocities of measured sheet flow- and rill flow well, without adjusting roughness parameters. This result alone indicates that flow accumulation in rills is important for simulating surface runoff and to explain micro-topographic adaptations as well as that the straightforward open book conceptualisation of the rill system is feasible.

Distribution of potential energy along the flow path for all simulations exhibited indeed a clear maximum. However, the maximum for the experiment with stronger and faster rill flow (lek_2) occurs farther upslope than for the experiment with

little to no rill flow (oek2_4). The Catflow-Rill simulation results show that for the rill domain the maximum potential energy is computed in close distance to flow Reynolds numbers in the range of 1800 to 2000 (Figure 15b). Although the model computes runoff with the Manning-Strickler formula, which is defined for turbulent flow, roughness coefficients were adjusted during the measurement campaign (Gerlinger, 1996) to fit into this framework, independent of prevalent flow regime and averaged over the experiment plot. It is therefore likely that for areas of the experiments with laminar flow, deduced Manning's

$n$ is underestimated. This effect becomes less important the more flow accumulates along the flow path and hydraulic conditions approach mixed to turbulent flow. With this in mind, the peak of total energy should occur some distance farther upstream than is shown in Figure 15 and therefore coincide for experiment lek_2 with the range of Reynolds numbers (~1500) where a flow regime transition is expected to occur (Emmett, 1970). This result indicates that laminar surface runoff is related to the downslope build-up of energy (energy added by flow accumulation > energy dissipated) and mixed/ turbulent flow to

net dissipation (energy added by flow accumulation < energy dissipated), highlighting that the distribution of potential energy in space is related to flow regime and therefore also erosional processes (see Figure 4).





Also, the reported critical values of *Re* by Emmett (1970) are higher than they are found by Phelps (1975) and were not discussed for influences of relative roughness $k_s = d_{sed}/d$. Phelps (1975) highlights that critical Reynolds numbers are lowered as relative roughness increases and he showed that critical *Re* can be as low as 400 for $k_s$=0.5. Therefore, we calculated

the relative roughness for the Weiherbach experiments (lek_2 and oek2_4) and found that $k_s$ stays for steady state conditions below 0.1. This justifies that the assumption of laminar to turbulent transition in the reported range by Emmett (1970) of approximately *Re*=1500 is valid for the here presented experiments.

These energy dynamics and the observed transition processes of flow regime and erosion fit well into our thermodynamic perspective. First, flow accumulation in rills shortens the flow paths with net increase of energy and lengthens the flow paths

with net decrease of energy. In line with Figure 3, our interpretation is that a hillslope which accumulates more flow in a rill is closer to a river system from an energetic point of view than a hillslope with little to no flow accumulation. Secondly, simulations of lek_2 with significant rill flow show that total kinetic energy of the rill flow domain approaches kinetic energy of the sheet flow domain in downslope direction. The same is observed for stream power distribution and is in line with maximum power principles, which state that maximum total power is achieved through equal distribution of power rates across

space and time (Kleidon, 2016). We see in fact strong similarities to the maximum power theorem in electrical circuits, which states that maximum power transfer is achieved if the load resistance matches the source resistance. For our example the source resistance would be represented by the resistance to flow within the sheet flow domain and the load resistance is equal to the resistance to flow of the rill flow domain. As sheet- and rill flow domains of Catflow-Rill share the same slope, not only ratio of resistance to flow, but also ratio of stream power of both domains approaches unity in downslope direction (see Annex C

for derivation). In this context we want to stress that the observed differences in rill- and sheet flow velocities (Scherer et al., 2012) were calibrated in Catflow-Rill using flow accumulation in rills only. We thus state that the irrigated hillslope stripe during steady state runoff approached a maximum power state in downslope flow direction as a unified system. We therefore believe that the separation of a planar surface into functional units of surface runoff is helpful to test these principles and find spatial equilibria, such as they have been proclaimed to exist e.g., between different hillslopes (Emmet, 1970) or at river

confluences (Howard, 1990). In this sense our results show that there might exist such an equilibrium between the sheet flow- and the rill flow domain, with the latter functioning as the highway that delivers sediment to the outlet and the sheet flow area operating as the sediment production source. If any one of the two functional units on average produces or delivers more sediment, there would be an imbalance, which undoubtedly will result in adaptation of energy gradients through erosion and deposition until balance between particle detachment and transport is once more achieved. The limit between particle

detachment and transport must be characterized by physical parameters such as the erosion resistance factor. In strictly fluvial systems such an equilibrium is expressed by the proportionality between discharge and loss of geopotential and in electrical circuits by Ohm's law (Rinaldo et al., 1998), an analogy which is also drawn to explain the development of natural systems through the maximum power theorem (Kleidon, 2016). From the calibrated simulation results of stream power and surface runoff velocity we computed average bed stresses per unit area and found that for experiment lek_2 the flow path length, where

forces in the rill domain significantly deviate from average erosion forces in the sheet flow domain, is where average bed stresses exceed measured critical erosion forces. Therefore, for a given rainfall-runoff event, with flow in structured rills and on unstructured inter-rill areas, rilling happens in accordance to average soil physical characteristics, which distributes dissipation equally between functional units. The initiation of a rill is a highly debated research topic, but studies show that rilling and channelization of land surfaces is related to the transition of diffusive erosion processes to an advective erosion

process (Smith and Bretherton, 1972; Tarboton et al., 1992). Smith (2010) argues that the advective transport is a result of water flow in direction of the energy gradient, but that diffusive transport is driven by topographic slope. In between he also distinguishes advection driven diffusive transport, which occurs when slope and energy gradient are not equal. Considering the resulting distributions of potential energy as shown in Figure 4 and in relative terms in Figure 3, we argue that formation of runoff, its flow regime and sediment transport mechanisms are directly related to the distribution of potential energy along





the flow path. A downslope increase of potential energy of the flow should therefore be related to diffusive sediment transport and predominantly laminar flow regime. Contrarily, a decrease of potential energy of water flow and an energy gradient approaching topographic slope is related to advective transport and therefore rill initiation, eventually transitioning into a fully turbulent flow regime and capacity limited sediment transport.

## 6 Summary and Conclusion

In this study we linked well known processes of Hortonian surface runoff (Shih and Yang, 2009) and thermodynamic principles (Kleidon, 2016). As the geomorphological development and surface runoff affect each other, we believe it to be particularly useful to account energy conversion rates for a unifying concept. This concept includes the conversion of free energy into heat and therefore enables us to state three hypotheses about the distribution of geopotential gradients which are the drivers of dissipation.

First, in our analysis we show that hillslopes as mass-accumulating systems are characterized by a distinctly different energetic behaviour in comparison to runon systems. The latter, e.g., a river does not necessarily lead to energetic maxima in space. When mass accumulation overweighs runon, which is usually the case for hillslopes, we will observe spatial maxima of potential energy. For these systems, a trade-off between mass gain and geopotential loss along a runoff flow path leads to energy maxima in space. Our results show that these maxima exist on hillslopes and therefore confirm hypothesis one. Then,
referring to our second hypothesis we interpret this finding as a result of the transitions of prevalent dissipation processes of Hortonian surface runoff. Hereby we present a reasonable theory why laminar flow regime should be related to sheet flow and mixed / turbulent flow is related to concentrated flow in rills and channels. For the presented experiments we show that the build-up of potential energy on hillslopes happens under laminar conditions, while decrease of potential energy along the flow path is a functionality of concentrated, turbulent rill flow. Hypothesis one and two feed into hypothesis three, which states that
dominance of surface runoff processes with respect to dissipation is directed to distribute potential energy uniformly along the flow path. This means that within each volume of hillslope surface the same amount of free energy converts into heat. This theory necessarily leads to maximum entropy (Leopold and Langbein, 1962; Kleidon, 2016) of the whole hillslope and for a mass accumulating system signifies that unit stream power is minimized along the flow path (Yang, 1976).

The spatial distribution of energy is therefore directed to evolve towards a state of maximum entropy and uniform energy
expenditure in space. A hillslope has several options to adjust the distribution of potential energy, which we distinguish into macro- and micro-topographic controls. For macro-topographic controls, an exponential distribution of geopotential and a converging width function leads to the largest flow path lengths with net free energy loss, which happens to be the prevalent three-dimensional hillslope form in nature and a result of long-term erosional adaptation (Kirkby, 1971). On a micro-topographic level, we analysed the formation of rills as the main adaptation mechanism to redistribute energy fluxes in space.
We show that the formation itself is not necessarily decreasing total frictional dissipation but rather serves as a means of faster flow accumulation, therefore distributing more energy to the transport of sediment within the rill and less energy to erosion by surface runoff. We found evidence that steady state runoff on the hillslopes developed to a maximum power state, where power is equally distributed in sheet and rill flow. In case this finding is corroborated within other experiments, it has important implications for constraining the control volume resistance of the rill system using optimality approaches.

**Author contribution**

S. Schroers conceptualized, implemented the CATFLOW-RILL extension, conducted the analysis and wrote the paper. O. Eiff conceptualized and supervised the hydraulic concepts. A. Kleidon reviewed and edited the thermodynamic concepts. U. Scherer provided the original CATFLOW setups and commented on surface runoff dynamics. J. Wienhöfer contributed to paper writing and CATFLOW modeling. E. Zehe oversaw the study and theory development as mentor.



**Competing interests**

The authors declare that they have no conflict of interest.

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





## Appendix A

### Energy flow between thermodynamic sub systems

For each $OTS_{sub}$ we apply Eq. (4) where potential and kinetic energy of the system do not change with time, so that:

$$0 = J_{f,net}^{pe}(x) + J_{f,net}^{ke}(x) + J_{Peff}^{pe}(x) - D_f(x) \tag{A1}$$

For potential energy conversion we obtain:

$$\frac{dE_f^{pe}(x)}{dt} = 0 = J_{f,net}^{pe}(x) + J_{Peff}^{pe}(x) - P_f(x) \tag{A2}$$

$$J_{f,net}^{pe}(x) + J_{Peff}^{pe}(x) = P_f(x)$$

While kinetic energy conversion is as follows:

$$\frac{dE_f^{ke}(x)}{dt} = 0 = P_f(x) - D_f(x) + J_{f,net}^{ke}(x) \tag{A3}$$

$$P_f(x) = D_f(x) - J_{f,net}^{ke}(x)$$

To relate the spatial distribution of energy with energy fluxes we recall that the downslope mass flux $\vec{v}$*m is associated with downslope flux of kinetic and potential energy. The net fluxes correspond to the divergence of the kinetic and potential energy flow. $J_f^{pe/ke}$ [watt] is here defined as the advective energy flux, which is the product of specific energy $E_{sp}$ [joule kg$^{-1}$] and flow rate $\rho * Q$ [kg s$^{-1}$]. As per definition of Eq. (4), $J_{f,net}$ is positive for a decrease of energy flux over the control volume and

therefore has the opposite sign to change in energy:

$$J_{f,net}^{pe/ke} = -div\left(J_f^{pe/ke}(x)\right) \tag{A4}$$

$$J_f^{pe} = E_{sp}^{pe}(x) * Q(x) = g * h(x) * \rho * Q(x) \tag{A5 a}$$

$$J_f^{ke} = E_{sp}^{ke}(x) * Q(x) = \frac{v(x)^2}{2} * \rho * Q(x) \tag{A5 b}$$

$$J_{Peff}^{pe}(x) = \rho * P_{eff}(x) * g * h(x) * b(x) \tag{A6}$$

Inserting the expressions for specific potential and kinetic energy (Eq. (A5) to Eq. (A6)) into Eq. (A2) and Eq. (A3), we get power (Eq. (A7)) and dissipation (Eq. (A8)) of flow energy per unit length in [W m$^{-1}$]:

$$P_f(x) = J_{f,net}^{pe}(x) + J_{Peff}^{pe}(x)$$

$$= \rho * g * \left(-\frac{dQ(x)}{dx} * h(x) - \frac{dh(x)}{dx} * Q(x) + P_{eff}(x) * h(x) * b(x)\right) \tag{A7}$$

$$D_f(x) = P_f(x) + J_{f,net}^{ke}(x)$$

$$= \rho * g * \left(-\frac{dQ(x)}{dx} * h(x) - \frac{dh(x)}{dx} * Q(x) + P_{eff}(x) * h(x) * b(x)\right)$$

$$-\frac{1}{2} * \rho * \left(\frac{dQ(x)}{dx} * v(x)^2 + 2 * v(x) * \frac{dv(x)}{dx} * Q(x)\right) \tag{A8}$$



**Appendix B**

**Run-off vs. run-on systems**

We now separate cases that accumulate runoff in downslope direction from cases with runon and transitory states (compare Fig. B1a, and Fig. B1b). In Figure 6b we already plotted $E_f^{pe}$ per unit length of all considered geopotential distributions z(x) with a constant width for a rainfall intensity of 50 mm hr$^{-1}$ without runon ($Q_0$=0). Here, Fig. B1a represents the transition from

a runoff-only to a runon-only system with a rainfall of 50 mm hr$^{-1}$ and runon of 20 kg s$^{-1}$, while Fig. B1b shows the potential energy distribution for a runon-only system without rainfall (I=0) and $Q_0$ of 20 kg s$^{-1}$.

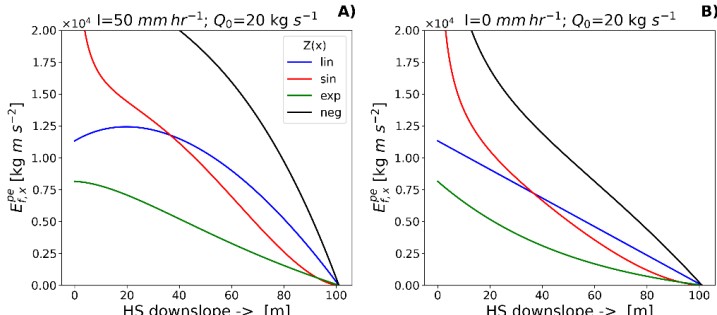

**Figure B1: Distribution of potential energy $E_f^{pe}$ per unit length in [Joule m$^{-1}$] a) with runon and runoff accumulation and c) with runon but without runoff accumulation**

From these calculations, it appears that runoff accumulating systems show distinct energy conversion dynamics in comparison to runon systems, where runon outplays rainfall accumulation. Most strikingly, the accumulation of runoff in space energetically counteracts the depletion of geopotential gradients and leads to an energetic maximum. This stands in contrast to pure runon systems where energy is continuously depleted downslope. In between the two extremes, we observe transitions, which can result in local maxima, such as in the example of a linear slope ($z_{lin}$(x), Fig. B1a) depending on rainfall and runon

intensities.

The unrealistically high potential energies for $z_{sin}$(x) and $z_{neg}$(x) (Fig. B1a, and Fig. B1b) within the first 20 meters of the hillslopes are due to our assumption that the energy gradient can be approximated by the slope of the terrain for the calculation of flow depth. The real energy gradient at these very small slopes cannot be approximated with slope and would require the solution of the momentum equation as e.g., in the shallow water equations in combination with proper upstream boundary

conditions but is out of scope for this analysis.





**Appendix C**


**Maximum Power in rill domain**

Flow on hillslope equivalent to current in circuit:

|  | Hillslope | Electrical Circuit |
|---|---|---|
| Flow | $Q = K * S^{0.5}$ | $I_{el} = \dfrac{1}{R_{el}} * V_{el}$ |
| Power | $P = Q^2 * \dfrac{1}{K} * \rho * g$ | $P_{el} = I_{el}^2 * R_{el}$ |

With

| symbol | unit | description |
|---|---|---|
| $I_{el}$ | [A] | Electrical current |
| $R_{el}$ | [Ω] | Resistance |
| $V_{el}$ | [V] | Voltage |
| $P_{el}$ | [W] | External power of the electrical circuit |
| $K$ | [m³ s⁻¹] | Conveyance of the channel: $K = \frac{1}{n} * A * R^{\frac{2}{3}}$ |
| $R_K$ | [m⁻³ s] | Resistance to flow: $R_K = 1/K$ |

Therefore, channel conveyance is the inverse of the resistance of the channel to transport flow.

If water is mainly falling on sheet flow area and flows therefore first on sheet-flow area with $R_K^{SF}$ and then accumulates in a channel with $R_K^{RF}$ the total resistance to flow is:

$$R_K = R_K^{SF} + R_K^{RF} \tag{C1}$$

Here we assume that $R_K^{SF}$ is fixed and that mainly resistance to flow of the rill adapts.

Total power in the rill is then:

$$P^{RF} = Q^2 * \frac{1}{R_K^{RF}} * \rho * g = \left( \left( R_K^{SF} + R_K^{RF} \right)^{-2} * S \right) * R_K^{RF} * \rho * g$$

$$= S * \rho * g \left( \underbrace{R_K^{RF} + 2 * R_K^{SF} + \frac{R_K^{SF\,2}}{R_K^{RF}}}_{T} \right)^{-1} \tag{C2}$$

C2 becomes maximum if the term "T" becomes minimum:

$$\frac{dT}{dR_K^{RF}} = 1 - \left( \frac{R_K^{SF}}{R_K^{RF}} \right)^2 \tag{C3}$$


The derivative (C3) becomes zero if:

$$R_K^{SF} = R_K^{RF}$$

Or equivalently:

$$K^{SF} = K^{RF}$$