# Peer review of "Morphological controls on surface runoff: An interpretation of steady-state energy patterns, maximum power states and dissipation regimes within a thermodynamic framework"

_Hydrology and Earth System Sciences, 2021_

## Referee Comment (RC1)

[referee-annotated manuscript omitted]

---

## Author Comment (AC2)

**Classification of dissipation regimes**

[Figure]

*Figure 1: 4-Box energy scheme of water and sediment flow*

For a flow that is comprised of water and sediment particles, the residual of the steady state energy balance can be calculated by combination of the energy balances of each medium. Potential energy of water is accumulated through rainfall, which converts into kinetic energy of water and heat (Eq. 1a). The magnitude of the dissipation of kinetic energy by viscous stress depends on flow regime (turbulent, laminar) as well as sediment transport. Sediment gains it's potential and kinetic energy from kinetic energy of the water, which is represented by the terms $P_{w,s}^{pe}$ and $P_{w,s}^{ke}$ (Eq. 1b). If these terms are positive, kinetic energy of water is converted into energy of sediment, which would be the case if water flow erodes the bed surface and picks up sediment particles. Reversely, surface runoff which already transports sediment might deposit some number of particles and sediments would lose potential and kinetic energy, leading to a negative sign of the terms $P_{w,s}^{pe}$ and $P_{w,s}^{ke}$.

Water energy balance:

$$\frac{dE_f^{pe}}{dt} = J_{f,net}^{pe} + J_{Peff}^{pe} - P_f \tag{1a}$$

$$\frac{dE_f^{ke}}{dt} = J_{f,net}^{ke} + P_f - D_f - P_{w,s}^{pe} - P_{w,s}^{ke} \tag{1b}$$

Sediment energy balance:

$$\frac{dE_s^{pe}}{dt} = J_{s,net}^{pe} - P_s + P_{w,s}^{pe} \tag{1c}$$

$$\frac{dE_s^{ke}}{dt} = J_{s,net}^{ke} + P_s - D_s + P_{w,s}^{ke} \tag{1d}$$

In steady state and with $P_s \approx D_s \approx 0$ (Ps represents the power of sediment to convert into movement of sediment and Ds is friction of grains against other grains) the residual of the water sediment energy balance is:

$$D_f = J_{f,net}^{pe} + J_{Peff}^{pe} + J_{s,net}^{pe} + J_{f,net}^{ke} + J_{s,net}^{ke} \tag{2}$$

Moreover, the flow regime might be of turbulent or laminar character, in the following indicated by superscripts T (turbulent flow regime) and L (laminar flow regime). With Eq. (2) We can now classify types of interactions

between water and sediment regarding the energy residual $D_f$. In the case of this simple two box scheme, we consider $D_f$ as dissipation of free energy into heat by the viscous stresses between water particles.

As a starting point, we define the following assumptions:

1) $J_{f,net}^{pe} + J_{Peff}^{pe} = P_f = constant$
2) Velocity does increase due to flow accumulation: $J_{f,net}^{ke} \leq 0$
3) Sediment moves with velocity of discharge: $v_{sed} = v_{water}$
4) Sediment is distributed homogeneously in the vertical of the water column (no vertical gradient of sediment concentration)
5) Kinetic energy divergence is much smaller than potential energy divergence:
$\left| J_{f,net}^{ke} \right| \ll \left| J_{f,net}^{pe} \right|$

First, we assume that the power of water to create kinetic energy is the same for each considered case (1) and second, we focus on cases where an influx of potential energy causes an increase in kinetic energy flow (2), combined with an increase of flow velocity. The sediment particles are assumed to move with the flow velocity (3) and are homogeneously distributed across the vertical of the water column (4). Also, most energy influx into the system is dissipated, meaning that the loss of geopotential energy is much larger than the gain of kinetic energy (5). These assumptions apply to a system which is not energy limited with large driving bed slopes and very shallow water depths.

In Figure 2 we plotted a flow chart diagram which separates the possible sediment water interactions. Starting at the top and continuing down a chart line the terms of the energy balance equation (Eq. 2) can be separated into different cases. Each line is named with a letter, resulting in a unique letter combination for each case. If no sediment is considered, the cases to distinguish (A) are just laminar and turbulent flow regime. The larger dissipative loss of turbulent flow leads to a slower flow velocity for turbulent flow than laminar flow and therefore $J_{f,net}^{ke,L} < J_{f,net}^{ke,T}$. All cases which include sediment transport start in the branch (B) of the flow chart. Branch (C) represents cases where $J_{Sed,net}^{pe} < 0$, which can only happen if sediment of the surface is eroded, and the total mass of sediment dissolved in the water flow is increasing. Further subdividing we find that the kinetic energy of sediment must increase as both $v_{Sed}$ and $m_{Sed}$ are increasing, meaning that $J_{Sed,net}^{ke}$ has to be smaller than zero, branch (E) is therefore not possible. If $J_{Sed,net}^{ke} < 0$ (branch (F)), we can again differentiate between laminar and turbulent flow regime. In contrast to cases with erosion only (C), branch (D) considers cases with deposition and erosion of sediment. The latter is however a special case as the sediment would have to lose more geopotential by slope difference than is gained by mass increase through erosion. Clarity exists if $J_{Sed,net}^{ke}$ is larger or equal to zero as this means that kinetic energy of sediment is decreasing, which can only be the case of deposition (mass decrease) as velocity increases (G). Contrarily, branch (H) is a collection of cases where the quantities of discharge, sediment concentration and magnitudes of slope and gradient determine whether sediment is eroded or deposited. These cases are either sediment mass decrease (deposition), which influences kinetic energy less than the increase of flow velocity, or little mass gain (erosion) in comparison to energy loss by slope. Both possibilities can be classified as little or no sediment mass change, therefore deposition and erosion being close to zero (H).

[Figure]

*Figure 2: Flowchart for dissipation ($D_f$) regimes*

The classification into dissipation regimes of water and sediment flow is summarized in Table 1. We can distinguish four principal cases: 1) Water and sediment flow with sediment mass increase (Erosion), 2) Water and sediment flow with little sediment mass in- or decrease, 3) Water flow without sediment transport, and 4) Water and sediment flow with sediment mass decrease (Deposition). Cases 2) and 3) might energetically be quite similar and could be considered a single case (No deposition and no erosion). Comparing the signs of the terms of Eq. 2 dissipation is lowest for erosion (case 1) and highest for deposition (case 3). For all cases turbulent flow is more dissipative than laminar flow ($v_w^L > v_w^T$ and $v_{Sed}^L > v_{Sed}^T$).

*Table 1: Dissipation regimes*

| Case | Route (Figure 2) | $J^{pe}_{f,net} + J^{pe}_{Peff,net}$ | $J^{pe}_{Sed,net}$ | $J^{ke}_{Sed,net}$ | $J^{ke}_{f,net}$ | $Erosion,$ $Deposition$: $\Delta m_{Sed}$ | Dissipation $D_f$ |
|------|------|------|------|------|------|------|------|
| 1 | B-C-F | Const. | < 0 | < 0 | < 0 | > 0 | Low |
| 2 | B-D-H | Const. | ≥ 0 | < 0 | < 0 | ~0 | |
| 3 | A | Const. | 0 | 0 | < 0 | 0 | |
| 4 | B-D-G | Const. | ≥ 0 | ≥ 0 | < 0 | < 0 | High |

*Table 1: Dissipation regimes*

---

## Author Response (AR1)

**HESS point-by-point responses to referees**

**1ˢᵗ referee K. Beven:**

K. Beven elaborated a detailed assessment of our study and presented a valuable critique, which we incorporated into the revised version of our manuscript.

We addressed his main comment regarding the limited evidence to our conclusions in two ways. First, we strengthened sect. 3 with an established theory, which links equilibrium hillslope forms and the dominant erosion process. This resulted in hillslope profiles which directly relate to the relative contribution of surface runoff and therefore allows a much more realistic analysis of the distribution of potential energy along the flow path. Second, we extended the analysis of rainfall simulation experiments in sect. 4 from 2 to 31 calibrated simulations and included a classification of steady state conditions, which are, as K. Beven commented as well, necessary for application of the presented steady state framework.

Related to these extensions is a restructuration of the theoretical section of the study (sect. 2), to which we added, as proposed by K. Beven a sub-section on the energy residual $D_f$. To this end, we elaborated on established friction laws and why constant parameters, such as Manning's $n$, are deemed to under- or overestimate friction losses and derived mean flow velocities of shallow surface runoff, especially in the case of high sediment loads. To avoid, as K. Beven mentioned, circularity of reasoning, we included an empirical friction law which implicitly incorporates heterogeneous friction losses and directly relates to equilibrium states of surface runoff on hillslopes. Finally, to draw focus on the accumulation and depletion of surface runoff potential energy as well as to streamline the narrative of the study, we excluded the comparison between surface runoff on hillslopes and in rivers.

Minor comments as addressed in K. Beven's commented manuscript are further addressed in the revised manuscript.

**Referee #2:**

Referee 2 made helpful comments regarding the structure and coherence of the study. We took advantage of the suggestions and focused the study on the accumulation and depletion of surface runoff potential energy exclusively on the hillslope-scale. Therefore, we excluded the comparison between surface runoff on hillslopes and in rivers and instead generally highlight the antagonistic effect of mass accumulation and geopotential loss.

To further streamline the manuscript, we restructured the introduction and highlighted the connection between the applications in sect. 3 and 4. In essence, sect. 3 relates geomorphological adaptations to surface runoff over long timescales, resulting in typical near equilibrium hillslope profiles, while in sect. 4 we analyze short-term morphological adaptations to surface runoff in the form of rills. In this second application we included in the revised manuscript the whole set of 31 evaluated experiments from Scherer (2008).

For both sections (sect. 3 and 4) we are able to show that adaptations of form, relative contribution of advective and diffusive processes, as well as the magnitude and location of the potential energy maximum are interrelated. In the revised manuscript we evaluated this finding by introducing a thermodynamic descriptor $D_f^{acc}/J_{in}^{acc}$ which normalizes the energy residual of a spatially integrated system for given influxes of energy.

Interestingly, a potential energy maximum farther upstream relates to hillslope profiles (sect. 3) as well as hillslope surfaces (sect. 4) which maximize this relative dissipation along the flow path, while at the same time maximizing relative export of kinetic energy.

Minor comments of referee 2 were also addressed in the revised manuscript, e.g., we followed the suggestion to exclude the wording "Hortonian", as we indeed do not focus on the runoff generation mechanism itself, but on overland flow in general.

---

## Referee Report (RR1)

This is a truly innovative paper aimed at linking hillslope surface flow, drainage structure formation, and erosion processes to Thermodynamics. The paper presents a new approach to understand hillslope processes by considering the omnipresent 1st and 2nd laws of thermodynamics. The paper is well readable, even though the concepts may be hard to grasp for the average hydrologist. The background is very well explained in 1.1.

Although there still is a lot to discover -- the authors have had to make a few simplifications to simplify their analysis -- I do consider this a landmark paper. While going through the paper, a few thoughts were triggered, which I would like to share with the authors, but which do not need to be addressed in a final version. I also found a few minor errors which I list at the end.

1. The authors consider steady state for obvious reasons of simplicity, but it makes me wonder. The hydrological environment is never in steady state; overland flow is always dynamic, and morphological change generally happens at extreme occurrences under transient conditions. Still, I agree with the authors that steady state energy patterns can inform us on the dominant processes of rill and channel formation and on the relation between resistance to flow (Manning's n), slope, hydraulic radius, flow width, sediment size and the bank full flow velocity. Can we reason why the steady state is informative? Maybe the maximum flow condition of an event, where time derivatives are also zero, is the dominant condition.

2. The approach hinges on the assumption that v=f(q). This has been found empirically, but there is also a logic in it. Let's consider a river along its convex geopotential trajectory from the hillslope to the sea. We see a gradually decreasing slope, a gradually decreasing bankfull flow velocity, decreasing transport capacity (a function of v^2.5), and -- as a result -- decreasing grain size of bed material. As the transport capacity reduces, the grain sizes that are too large to transport have to be dumped over and on the banks by exceptional flood events. The channel is thus a transporter of sediment of a certain grainsize belonging to a certain longitudinal coordinate in the channel, with a specific slope and bankfull flow velocity. To maintain a stable channel, the minimum flow velocity corresponds with the transport capacity required for this sediment and with minimum energy expenditure. Hence, there is a thermodynamic connection between roughness, slope, hydraulic roughness, flow velocity and sediment size, which we recognize in the empirical equations for resistance to flow and in the cited equation that v=f(q), implying that on a hillslope with uniform sediment particles, there is an additional relation between flow velocity, roughness, and slope, resulting in a constant velocity and constant transport capacity. There must be a thermodynamic explanation for this empirical relation. It brings to my mind Riggs' equation for open channel flow, where he found that Manning's roughness is a function of slope S, hydraulic radius and cross-sectional area A, leading to a simple function where Q=f(A,S), which performed equally food as the Manning equation where experienced hydrologists estimated Manning's n.

3. In Fig.5 the advective erosion (SW) is closest to uniform relative dissipation. In my view the convergent and advective process is the closest to the natural form of a drainage

network. Would it then be safe to assume that the uniform relative dissipation is the thermodynamic optimum for drainage networks (you observe this as well in L439-442), and that we can use this as the basic assumption for investigating the issues mentioned here under 2?

Some small errors:
L380: "is" to replace "as"
L395: I think you switched b) and c) in the caption.

Reference:
Riggs, H.C., 1976. A simplified slope-area method for estimating flood discharges in natural channels. Journal Research U.S. Geol. Survey Vol. 4, No. 3.